# Concave Utility Reinforcement Learning with Zero-Constraint Violations

**Mridul Agarwal**                                                  *agarw180@purdue.edu*
*Purdue University*

**Qinbo Bai**                                                       *bai113@purdue.edu*
*Purdue University*

**Vaneet Aggarwal**                                                 *vaneet@purdue.edu*
*Purdue University*

**Reviewed on OpenReview:** *https://openreview.net/forum?id=WXVkgkPXRk*

## Abstract

We consider the problem of tabular infinite horizon concave utility reinforcement learning (CURL) with convex constraints. For this, we propose a model-based learning algorithm that also achieves zero constraint violations. Assuming that the concave objective and the convex constraints have a solution interior to the set of feasible occupation measures, we solve a tighter optimization problem to ensure that the constraints are never violated despite the imprecise model knowledge and model stochasticity. We use Bellman error-based analysis for tabular infinite-horizon setups which allows analyzing stochastic policies. Combining the Bellman error-based analysis and tighter optimization equation, for $T$ interactions with the environment, we obtain a high-probability regret guarantee for objective which grows as $\tilde{O}(1/\sqrt{T})$, excluding other factors. The proposed method can be applied for optimistic algorithms to obtain high-probability regret bounds and also be used for posterior sampling algorithms to obtain a loose Bayesian regret bounds but with significant improvement in computational complexity.

## 1 Introduction

In many applications where a learning agent uses reinforcement learning to find optimal policies, the agent optimizes a concave function of the expected rewards or the agent must satisfy certain constraints while maximizing an objective (Altman & Schwartz, 1991; Roijers et al., 2013). For example, in network scheduling, a controller can maximize fairness of the users using a concave function of the average reward of each of the users (Chen et al., 2021). Consider a scheduler which allocates a resource to users. Each user obtains some reward based on their current state. The goal of the scheduler is to maximize fairness among the users. However, there are certain preferred users for which some service level agreements (SLA) must be made. For this setup, the scheduler aims to find a policy which maximizes the fairness while ensuring the SLA constraints of the preferred users are met. Note that, here, the objective is a non-linear concave utility in the presence of constraints on service level agreement. Setups with constraints also exist in autonomous vehicles where the goal is to reach the destination quickly while ensuring the safety of the surroundings (Le et al., 2019; Tessler et al., 2018). Further, an agent may aim to efficiently explore the environment by maximizing the entropy, which is a concave function of the distribution induced over the state and action space (Hazan et al., 2019).

Owing to the variety of the use cases, recently, there has been significant effort to make RL algorithms for setups with constraints, or concave utilties, or both. For episodic setup, works range from model based algorithms (Brantley et al., 2020; Yu et al., 2021) to primal-dual based model-free algorithms (Ding et al., 2021). Recently, there has been a thrust towards developing algorithms which can also achieve zero-constraint violations in the learning phase as well (Wei et al., 2022a; Liu et al., 2021; Bai et al., 2022b). However, for

the episodic setup, the majority of the current works consider the weaker regret definition specified by Efroni et al. (2020) and only achieve zero expected constraint violations. Further, these algorithms require the knowledge of a safe policy following which the agent does not violate constraints, or the knowledge of the Slater's gap $\delta$ which determines how far a safe policy is from the constraint boundary.

The definition which considers the average over time makes sense for an infinite horizon setup as the long-term average is naturally defined (Puterman, 2014). For a tabular infinite-horizon setup, Singh et al. (2020) proposed an optimistic epoch-based algorithm. Much recently, Chen et al. (2022) proposed an Optimistic Online Mirror Descent based algorithm. In this work, we consider the problem of maximizing concave utility of the expected rewards while also ensuring that a set of convex constraints of the expected rewards are also satisfied. Moreover, we aim to develop algorithms that can also ensure that the constraints are not violated during the training phase as well. We work with tabular MDP with infinite horizon. For such setup, our algorithm updates policies as it learns the system model. Further, our approach also bounds the accumulated observed constraint violations as compared to the expected constraint violations.

For infinite horizon setups for non-constrained setup, the regret analysis has been widely studied (Fruit et al., 2018; Jaksch et al., 2010). However, we note that the dealing with constraints and non-linear setup requires additional attention because of the stochastic policies. Further, unlike episodic setup, the distribution at the epoch is not constant and hence the policy switching cost has to be accounted explicitly. Prior works in infinite horizon also faced this issue and provide some tools to overcome this limitation. Singh et al. (2020) builds confidence intervals for transition probability for every next state given the current state-action pair and obtains a regret bound of $O(T^{2/3})$. Chen et al. (2022) obtains a regret bound of $O(T_M\sqrt{T})$ with $O(T_M^2 S^3 A)$ constraint violations for ergodic MDPs with $T_M$ mixing time following an analysis which works with confidence intervals on both transition probability vectors and value functions.

To overcome the limitations mentioned in previous analysis and to obtain a tighter result, we propose an optimism based UC-CURL algorithm which proceeds in epochs $e$. At each epoch, we solve for an policy which considers constraints tighter by $\epsilon_e$ than the true bounds for the optimistic MDP in the confidence intervals for the transition probabilities. Further, as the knowledge of the model improves with increased interactions with the environment, we reduce this tightness. This $\epsilon_e$-sequence is critical to our algorithm as, if the sequence decays too fast, the constraints violations cannot be bounded by zero. If this sequence decays too slow, the objective regret may not decay fast enough. Further, using the $\epsilon_e$-sequence, we do not require the knowledge of the total time $T$ for which the algorithm runs.

We bound our regret by bounding the gap between the optimal policy in the feasible region and the optimal policy for the optimization problem with $\epsilon_e$ tight constraints. We bound this gap with a multiplicative factor of $O(1/\delta)$, where $\delta$ is Slater's parameter. Based on our analysis using the Slater's parameter $\delta$, we consider a case where a lower bound $T_l$ on the time horizon $T$ is known. This knowledge of $T_l$ allows us to relax our assumption on $\delta$.

Further, for the regret analysis of the proposed UC-CURL algorithm, we use Bellman error for infinite horizon setup to bound the difference between the performance of optimistic policy on the optimistic MDP and the true MDP. Compared to analysis of Jaksch et al. (2010), this allows us to work with stochastic policies. We bound our regret as $\tilde{O}(\frac{1}{\delta}LdT_M S\sqrt{A/T} + CT_M S^2 A/T(1-\rho))$ and constraint violations as 0, where $S$ and $A$ are the number of states and actions respectively, $L$ is the Lipschitz constant of the objective and constraint functions, $d$ is the number of costs the agent is trying to optimize, and $T_M$ is the mixing time of the MDP. The Bellman error based analysis along with Slater's slackness assumption also allows to develop posterior sampling based methods for constrained RL (see Appendix G) by showing feasibility of the optimization problem for the sampled MDPs.

To summarize our contributions, we improve prior results on infinite horizon concave utility reinforcement learning setup on multiple fronts. First, we consider convex function for objectives and constraints. Second, even with a non-linear function setup, we reduce the regret order to $O(T_M S\sqrt{A/T})$ and bound the constraint violations with 0. Third, our algorithm does not require the knowledge of the time horizon $T$, safe policy, or Slater's gap $\delta$. Finally, we provide analysis for posterior sampling algorithm which improves both empirical performance and computational complexity.

## 2  Related Works

**Constrained RL:** Altman (1999) builds the formulation for constrained MDPs to study constrained reinforcement learning and provides algorithms for obtaining policies with known transition models. Zheng & Ratliff (2020) considered an episodic CMDP (Constrained Markov Decision Processes) and use an optimism based algorithm to bound the constraint violation as $\tilde{O}(1/T^{0.25})$ with high probability. Kalagarla et al. (2021) also considered the episodic setup to obtain PAC-style bound for an optimism based algorithm. Ding et al. (2021) considered the setup of $H$-episode length episodic CMDPs with $d$-dimensional linear function approximation to bound the constraint violations as $\tilde{O}(d\sqrt{H^5/T})$ by mixing the optimal policy with an exploration policy. Efroni et al. (2020) proposes a linear-programming and primal-dual policy optimization algorithm to bound the regret as $O(S\sqrt{H^3/T})$. Wei et al. (2022a); Liu et al. (2021) considered the problem of ensuring zero constraint violations using a model-free algorithm for tabular MDPs with linear rewards and constraints. However, for infinite horizon setups, the analysis from finite horizon algorithms does not directly hold. This is because finite horizon setups can update the policy after every episode. But this policy switch modifies the induced Markov chains which takes time to converge to a stationary distribution.

Xu et al. (2021) considered an infinite horizon discounted setup with constraints and obtain global convergence using policy gradient algorithms. Bai et al. (2022b) proposed a conservative stochastic model-free primal-dual algorithm for infinite horizon discounted setup. Ding et al. (2020); Bai et al. (2023) also considered an infinite horizon discounted setup with parametrization. They used a natural policy gradient to update the primal variable and sub-gradient descent to update the dual variable. In addition to the above results on discounted MDPs, the long-term rewards have also been considered. Singh et al. (2020) considered the setup of infinite-horizon ergodic CMDPs with long-term average constraints with an optimism based algorithm. Gattami et al. (2021) analyzed the asymptotic performance for Lagrangian based algorithms for infinite-horizon long-term average constraints, however they only show convergence guarantees without explicit convergence rates. Chen et al. (2022) provided an optimistic online mirror descent algorithm for ergodic MDPs which obtain a regret bound of $O(T_M S\sqrt{SAT})$, and Wei et al. (2022b) provided a model free SARSA algorithm which obtains a regret bound of $O(\sqrt{SA}T^{5/6})$ for constrained MDPs. Agarwal et al. (2022b) proposed a posterior sampling based algorithm for infinite horizon setup with a regret of $\tilde{O}(T_M S\sqrt{AT})$ and constraint violation of $\tilde{O}(T_M S\sqrt{AT})$.

| Algorithm(s) | Setup | Regret | Constraint Violation | Non-Linear |
|---|---|---|---|---|
| CONRL (Brantley et al., 2020) | FH | $\tilde{O}(LH^{5/2}S\sqrt{A/K})$ | $O(H^{5/2}S\sqrt{A/K})$ | Yes |
| MOMA (Yu et al., 2021) | FH | $\tilde{O}(LH^{3/2}\sqrt{SA/K})$ | $\tilde{O}(H^{3/2}\sqrt{SA/K})$ | Yes |
| TripleQ (Wei et al., 2022a) | FH | $\tilde{O}(\frac{1}{\delta}H^4\sqrt{SA}K^{-1/5})$ | 0 | No |
| OptPess-LP (Liu et al., 2021) | FH | $\tilde{O}(\frac{H^3}{\delta}\sqrt{S^3A/K})$ | 0 | No |
| OptPess-Primal Dual (Liu et al., 2021) | FH | $\tilde{O}(\frac{H^3}{\delta}\sqrt{S^3A/K})$ | $\tilde{O}(H^4S^2A/\delta)$ | No |
| UCRL-CMDP (Singh et al., 2020) | IH | $\tilde{O}(\sqrt{SA}T^{-1/3})$ | $\tilde{O}(\sqrt{SA}/T^{1/3})$ | No |
| Chen et al. (Chen et al., 2022) | IH | $\tilde{O}(\frac{1}{\delta}T_M S\sqrt{SA/T})$ | $\tilde{O}(\frac{1}{\delta^2}T_M^2 S^3 A)$ | No |
| Wei et al. (Wei et al., 2022b) | IH | $\tilde{O}(\frac{1}{\delta}\sqrt{SA}T^{-1/6})$ | 0 | No |
| Agarwal et al. (Agarwal et al., 2022b) | IH | $\tilde{O}(T_M S\sqrt{A/T})$ | $\tilde{O}(T_M S\sqrt{A/T})$ | No |
| UC-CURL (This work) | IH | $\tilde{O}(\frac{1}{\delta}LT_M S\sqrt{A/T})$ | 0 | **Yes** |

Table 1: Overview of work for constrained reinforcement learning setups. For finite horizon (FH) setups, $H$ is the episode length and $K$ is the number of episodes for which the algorithm runs. For infinite horizon (IH) setups, $T_M$ denotes the mixing time of the MDP, and $T$ is the time for which algorithm runs. $L$ is the Lipschitz constant. We note that all the IH setups assume ergodic MDPs, where the FH setups do not require the ergodic assumption as the system resets to the final state after every episode.

**Concave Utility RL:** Another major research area related to this work is concave utility RL (Hazan et al., 2019). Along this direction, Cheung (2019) considered a concave function of expected per-step vector reward and developed an algorithm using Frank-Wolfe gradient of the concave function for tabular infinite horizon MDPs. Agarwal & Aggarwal (2022); Agarwal et al. (2022a) also considered the same setup using a posterior sampling based algorithm. Recently, Brantley et al. (2020) combined concave utility reinforcement learning and constrained reinforcement learning for an episodic setup. Yu et al. (2021) also considered the case of

episodic setup with concave utility RL. However, both (Brantley et al., 2020) and (Yu et al., 2021) consider the weaker regret definition by Efroni et al. (2020), and Cheung (2019); Yu et al. (2021) do not target the convergence of the policy. Further, these works do not target zero-constraint violations. Recently, policy gradient based algorithms have also been studied for discounted infinite horizon setup (Bai et al., 2022a).

Another parallel line of work in RL which deals with concave utilities is variational policy gradient (Zhang et al., 2021; 2020). However, they consider discounted MDPs whereas we consider undiscounted setup for our work.

Compared to prior works, we consider the constrained reinforcement learning with convex constraints and concave objective function. Using infinite-horizon setup, we consider the tightest possible regret definition. Further, we achieve zero constraint violations with objective regret tight in $T$ using an optimization problem with decaying tightness. A comparative survey of key prior works and our work is also presented in Table 1.

## 3 Problem Formulation

We consider an ergodic tabular infinite-horizon constrained Markov Decision Process $\mathcal{M} = (\mathcal{S}, \mathcal{A}, r, f, c_1, \cdots, c_d, g, P, \phi)$. $\mathcal{S}$ is finite set of $S$ states, and $\mathcal{A}$ is a finite set of $A$ actions. $P : \mathcal{S} \times \mathcal{A} \to \Delta(\mathcal{S})$ denotes the transition probability distribution such that on taking action $a \in \mathcal{A}$ in state $s \in \mathcal{S}$, the system moves to state $s' \in \mathcal{S}$ with probability $P(s'|s, a)$. $r : \mathcal{S} \times \mathcal{A} \to [0, 1]$ and $c_i : \mathcal{S} \times \mathcal{A} \to [0, 1], i \in 1, \cdots, d$ denotes the average reward obtained and average costs incurred in state action pair $(s, a) \in \mathcal{S} \times \mathcal{A}$, and $\phi$ is the distribution over the initial state.

The agent interacts with $\mathcal{M}$ in time-steps $t \in 1, 2, \cdots$ for a total of $T$ time-steps. We note that $T$ is possibly unknown and $s_1 \sim \phi$. At each time $t$, the agent observes state $s_t$ and plays action $a_t$. The agent selects an action on observing the state $s$ using a policy $\pi : \mathcal{S} \to \Delta(\mathcal{A})$, where $\Delta(\mathcal{A})$ is the probability simplex on the action space. On following a policy $\pi$, the long-term average reward of the agent is denoted as:

$$\lambda_\pi^P = \lim_{\tau \to \infty} \mathbb{E}_{\pi, P} \left[ \sum\nolimits_{t=1}^{\tau} r(s_t, a_t)/\tau \right] \tag{1}$$

where $\mathbb{E}_{\pi, P}[\cdot]$ denotes the expectation over the state and action trajectory generated from following $\pi$ on transitions $P$. The long-term average reward can also be represented as:

$$\lambda_\pi^P = \sum\nolimits_{s,a} \rho_\pi^P(s, a) r(s, a) = \lim_{\gamma \to 1} (1 - \gamma) V_\gamma^{\pi, P}(s) \ \forall s \in \mathcal{S}$$

where $V_\gamma^{\pi, P}(s)$ is the discounted cumulative reward on following policy $\pi$, and $\rho_\pi^P \in \Delta(\mathcal{S} \times \mathcal{A})$ is the steady-state occupancy measure generated from following policy $\pi$ on MDP with transitions $P$ (Puterman, 2014). Similarly, we also define the long-term average costs as follows:

$$\zeta_\pi^P(i) = \lim_{\tau \to \infty} \mathbb{E}_{\pi, P} \left[ \sum\nolimits_{t=1}^{\tau} c_i(s_t, a_t)/\tau \right] = \lim_{\gamma \to 1} (1 - \gamma) V_\gamma^{\pi, P}(s; i) \quad \forall s \in \mathcal{S}$$
$$= \sum\nolimits_{s,a} \rho_\pi^P(s, a) c_i(s, a) \tag{2}$$

The agent interacts with the CMDP $\mathcal{M}$ for $T$ time-steps in an online manner and aims to maximize a function $f : [0, 1] \to \mathbb{R}$ of the average per-step reward. Further, the agent attempts to ensure that a function of average per-step costs $g : [0, 1]^d \to \mathbb{R}$ is at most 0. In the hindsight, the agents wants to play a policy $\pi^*$ which which satisfies:

$$\max_\pi f\left(\lambda_\pi^P\right) \quad s.t. \quad g\left(\zeta_\pi^P(1), \cdots, \zeta_\pi^P(d)\right) \leq 0 \tag{3}$$

Let $P_{\pi,s}^t = \prod_{t'=1}^{t} P_\pi$ denote the $t$-step transition probability on following policy $\pi$ in MDP $\mathcal{M}$ starting from some state $s$ where $P_\pi(\cdot|s) = \sum_a \pi(a|s) P(\cdot|s, a)$. Let $T_{s \to s'}^\pi$ denote the time taken by the Markov chain induced by the policy $\pi$ to hit state $s'$ starting from state $s$. Further, let $T_M := \max_\pi \mathbb{E}[T_{s \to s'}^\pi]$ be the mixing time of the MDP $\mathcal{M}$. We now introduce our assumptions on the MDP $\mathcal{M}$.

**Assumption 3.1.** For MDP $\mathcal{M}$, we have $\|P_{\pi,s}^t - P_\pi\| \leq C\rho^t$ with $P_\pi$ being the long-term steady state distribution induced by policy $\pi$, and $C > 0$ and $\rho < 1$ are problem specific constants. Additionally, the mixing time of the MDP $\mathcal{M}$ if finite or $T_M < \infty$. In other words, the MDP, $\mathcal{M}$, is ergodic.

**Assumption 3.2.** The rewards $r(s,a)$, the costs $c_i(s,a); \forall\, i$, and the functions $f$ and $g$ are known to the agent.

**Assumption 3.3.** The scalarization function $f$ is jointly concave and the constraints $g$ are jointly convex. Hence for any arbitrary distributions $\mathcal{D}_1$ and $\mathcal{D}_2$, the following holds.

$$f\left(\mathbb{E}_{x\sim\mathcal{D}_1}[x]\right) \geq \mathbb{E}_{x\sim\mathcal{D}_1}[f(x)] \tag{4}$$

$$g\left(\mathbb{E}_{\mathbf{x}\sim\mathcal{D}_2}[\mathbf{x}]\right) \leq \mathbb{E}_{\mathbf{x}\sim\mathcal{D}_2}[g(\mathbf{x})]; \; \mathbf{x} \in \mathbb{R}^d \tag{5}$$

**Assumption 3.4.** The function $f$ and $g$ are assumed to be a $L-$ Lipschitz function, or

$$|f(x) - f(y)| \leq L|x - y|; \; x, y \in \mathbb{R} \tag{6}$$

$$|g(\mathbf{x}) - g(\mathbf{y})| \leq L\|\mathbf{x} - \mathbf{y}\|_1; \; \mathbf{x}, \mathbf{y} \in \mathbb{R}^d \tag{7}$$

*Remark* 3.5. We consider a standard setup of concave and the Lipschitz function as considered by Cheung (2019); Brantley et al. (2020); Yu et al. (2021). Note that the analysis in this paper directly works for $f : \mathbb{R}^K \to \mathbb{R}$, where the function takes as input $K$ average per-step rewards for $K$ objectives.

*Remark* 3.6. For non-Lipshitz continuous functions such as entropy, we can obtain maximum entropy exploration if choose function $f = -\sum_k \lambda_k \log(\lambda_k + \eta)$ with $r_k(s,a) = \mathbf{1}_{\{s_k,a_k\}}$ for a particular state action pair $s_k, a_k$ and choosing $K = S \times A$ to cover all state-action pairs and a regularizer $\eta$ (Hazan et al., 2019).

**Assumption 3.7.** There exists a policy $\pi$, and one constant $\delta > LdST_M\sqrt{(A\log T)/T} + (CSA\log T)/(T(1-\rho))$ such that

$$g\left(\zeta_\pi^P(1), \cdots, \zeta_\pi^P(d)\right) \leq -\delta \tag{8}$$

This assumption is again a standard assumption in the constrained RL literature (Efroni et al., 2020; Ding et al., 2021; 2020; Wei et al., 2022a). $\delta$ is referred as Slater's constant. Ding et al. (2021) assumes that the Slater's constant $\delta$ is known. Wei et al. (2022a) assumes that the number of iterations of the algorithm is at least $\tilde{\Omega}(SAH/\delta)^5$ for episode length $H$. On the contrary, we simply assume the existence of $\delta$ and a lower bound on the value of $\delta$ which gets relaxed as the agent acquires more time to interact with the environment.

Any online algorithm starting with no prior knowledge will need to obtain estimates of transition probabilities $P$ and obtain reward $r$ and costs $c_k, \forall\, k \in \{1, \cdots, d\}$, for each state action pair. Initially, when algorithm does not have good estimate of the model, it accumulates a regret as well as violates constraints as it does not know the optimal policy. We define reward regret $R(T)$ as the difference between the average cumulative reward obtained vs the expected rewards from running the optimal policy $\pi^*$ for $T$ steps, or

$$R(T) = f\left(\lambda_{\pi^*}^P\right) - f\left(\sum_{t=1}^T r(s_t, a_t)/T\right)$$

Additionally, we define constraint regret $C(T)$ as the gap between the constraint function and incurred and constraint bounds, or

$$C(T) = \left(g\left(\sum_{t=1}^T c_1(s_t, a_t)/T, \cdots, \sum_{t=1}^T c_d(s_t, a_t)/T\right)\right)_+, \; \text{where} (x)_+ = \max(0, x)$$

In the following section, we present a model-based algorithm to obtain this policy $\pi^*$, and reward regret and the constraint regret accumulated by the algorithm.

## 4   Algorithm

We now present our algorithm UC-CURL and the key ideas used in designing the algorithm. Note that if the agent is aware of the true transition $P$, it can solve the following optimization problem for the optimal

feasible policy.

$$\max_{\rho(s,a)} f\big(\sum_{s,a} r(s,a)\rho(s,a)\big) \tag{9}$$

with the following set of constraints,

$$\sum_{s,a} \rho(s,a) = 1, \quad \rho(s,a) \geq 0 \tag{10}$$

$$\sum_{a \in \mathcal{A}} \rho(s',a) = \sum_{s,a} P(s'|s,a)\rho(s,a) \tag{11}$$

$$g\big(\sum_{s,a} c_1(s,a)\rho(s,a), \cdots, \sum_{s,a} c_d(s,a)\rho(s,a)\big) \leq 0 \tag{12}$$

for all $s' \in \mathcal{S}$, $\forall s \in \mathcal{S}$, and $\forall a \in \mathcal{A}$. Equation (11) denotes the constraint on the transition structure for the underlying Markov Process. Equation (10) ensures that the solution is a valid probability distribution. Finally, Equation (12) are the constraints for the constrained MDP setup which the policy must satisfy. Using the solution for $\rho$, we can obtain the optimal policy as:

$$\pi^*(a|s) = \frac{\rho(s,a)}{\sum_{b \in \mathcal{A}} \rho(s,b)} \forall\ s,a \tag{13}$$

However, the agent does not have the knowledge of $P$ to solve this optimization problem, and thus starts learning the transitions with an arbitrary policy. We first note that if the agent does not have complete knowledge of the transition $P$ of the true MDP $\mathcal{M}$, it should be conservative in its policy to allow room to violate constraints. Based on this idea, we formulate the $\epsilon$-tight optimization problem by modifying the constraint in Equation (12) as.

$$g\big(\sum_{s,a} c_1(s,a)\rho_\epsilon(s,a), \cdots, \sum_{s,a} c_d(s,a)\rho_\epsilon(s,a)\big) \leq -\epsilon \tag{14}$$

Let $\rho_\epsilon$ be the solution of the $\epsilon$-tight optimization problem, then the optimal conservative policy becomes:

$$\pi_\epsilon^*(a|s) = \frac{\rho_\epsilon(s,a)}{\sum_{b \in \mathcal{A}} \rho_\epsilon(s,b)} \forall\ s,a \tag{15}$$

We are now ready to design our algorithm UC-CURL which is based on the optimism principle (Jaksch et al., 2010). The UC-CURL algorithm is presented in Algorithm 1. The algorithm proceeds in epochs $e$. The algorithm maintains three key variables $\nu_e(s,a)$, $N_e(s,a)$, and $\hat{P}(s,a,s')$ for all $s,a$. $\nu_e(s,a)$ stores the number of times state-action pair $(s,a)$ are visited in epoch $e$. $N_e(s,a)$ stores the number of times $(s,a)$ are visited till the start of epoch $e$. $\hat{P}(s,a,s')$ stores the number of times the system transitions to state $s'$ after taking action $a$ in state $s$. Another key parameter of the algorithm is $\epsilon_e = K\sqrt{(\log t_e)/t_e}$ where $t_e$ is the start time of the epoch $e$ and $K$ is a configurable constant. Using these variables, the agent solves for the optimal $\epsilon_e$-conservative policy for the optimistic MDP by replacing the constraints in Equation (11) by:

$$\sum_{a \in \mathcal{A}} \rho(s',a) \leq \sum_{s,a} \tilde{P}_e(s'|s,a)\rho(s,a) \tag{16}$$

$$\tilde{P}_e(s'|s,a) > 0, \sum_{s'} \tilde{P}_e(s'|s,a) = 1 \tag{17}$$

$$\|\tilde{P}_e(\cdot|s,a) - \frac{\hat{P}(s,a,\cdot)}{1 \vee N_e(s,a)}\|_1 \leq \sqrt{\frac{14S\log(2At)}{1 \vee N_e(s,a)}} \tag{18}$$

for all $s' \in \mathcal{S}, \forall s \in \mathcal{S}$, and $\forall a \in \mathcal{A}$ and $x \vee y = \max(x,y)$. Equation (18) ensures that the agent searches for optimistic policy in the confidence intervals of the transition probability estimates.

Combining the right hand side of (16) with (10) gives

$$\sum_{s'} \sum_{s,a} \tilde{P}_e(s'|s,a)\rho(s,a) = 1 = \sum_{s',a} \rho(s',a)$$

---

**Algorithm 1** UC-CURL

---

**Parameters**: $K$

**Input**: $S$, $A$, $r$, $d$, $c_i \ \forall \ i \in [d]$

1: Let $t = 1$, $e = 1$, $\epsilon_e = K\sqrt{\frac{\ln t}{t}}$
2: **for** $(s, a) \in \mathcal{S} \times \mathcal{A}$ **do**
3: $\quad \nu_e(s, a) = 0, N_e(s, a) = 0, \widehat{P}(s', a, s) = 0 \forall \ s' \in \mathcal{S}$
4: **end for**
5: Solve for policy $\pi_e$ using Eq. (19)
6: **for** $t \in \{1, 2, \cdots\}$ **do**
7: $\quad$ Observe $s_t$, and play $a_t \sim \pi_e(\cdot|s_t)$
8: $\quad$ Observe $s_{t+1}$, $r(s_t, a_t)$ and $c_i(s_t, a_t) \ \forall \ i \in [d]$
9: $\quad \nu_e(s_t, a_t) = \nu_e(s_t, a_t) + 1$
10: $\quad \widehat{P}(s_t, a_t, s_{t+1}) = \widehat{P}(s_t, a_t, s_{t+1}) + 1$
11: $\quad$ **if** $\nu_e(s, a) = \max\{1, N_e(s, a)\}$ for any $s, a$ **then**
12: $\qquad$ **for** $(s, a) \in \mathcal{S} \times \mathcal{A}$ **do**
13: $\qquad\quad N_{e+1}(s, a) = N_e(s, a) + \nu_e(s, a)$
14: $\qquad\quad e = e + 1, \nu_e(s, a) = 0$
15: $\qquad$ **end for**
16: $\qquad \epsilon_e = K\sqrt{\frac{\ln t}{t}}$
17: $\qquad$ Solve for policy $\pi_e$ using Eq. (19)
18: $\quad$ **end if**
19: **end for**

---

Thus, joint with (16), we see that equality in (16) will be satisfied at the boundary as $\sum_a \rho(s', a)$ for some $s'$ can never exceed the boundary to compensate for another $s'$ and hence, for all $s'$, $\sum_a \rho(s', a)$ will lie on the boundary. In other words, the above constraints give $\sum_{a \in \mathcal{A}} \rho(s', a) = \sum_{s,a} \tilde{P}_e(s'|s, a)\rho(s, a)$. Further, we note that the region for the constraints is convex. This is because the set $\{x, y, z : xy \geq z\}$ is convex when $x, y, z \geq 0$. We note that even though the optimization problem may look non-convex due to constraints having product of two variables, we see Equations (9), (14), and (16)-(18) form a convex optimization problem. We expand more on this in Appendix B. We note that (Rosenberg & Mansour, 2019) provide another approach to obtain a convex optimization problem for optimistic MDP.

Let $\rho_e$ be the solution for $\epsilon_e$-tight optimization equation for the optimistic MDP. Then, we obtain the optimal conservative policy for epoch $e$ as:

$$\pi_e(a|s) = \frac{\rho_e(s, a)}{\sum_{b \in \mathcal{A}} \rho_e(s, b)} \forall \ s, a \tag{19}$$

The agent plays the optimistic conservative policy $\pi_e$ for epoch $e$. Note that the conservative parameter $\epsilon_e$ decays with time. As the agent interacts with the environment, the system model improves and the agent does not need to be as conservative as before. This allows us to bound both constraint violations and the objective regret. Further, if during the initial iterations of the algorithms a conservative solution is not feasible, we can ignore the constraints completely. We will show that the conservation behavior is required when $t = \Theta(T)$ to compensate for the violations in the initial period of the algorithm E.2.

For the UC-CURL algorithm described in Algorithm 1, we choose $\{\epsilon_e\} = \{K\sqrt{(\log t_e)/t_e}\}$. However, if the agent has access to a lower bound $T_l$ (Assumption 3.7) on the time horizon $T$, the algorithm can change the $\epsilon_e = K\sqrt{(\ln(t_e \vee T_l))/(t_e \vee T_l)} \leq \delta$ in each epoch $e$ as follows. Note that if $T_l = 0$, $\epsilon_e$ becomes as specified in Algorithm 1 and if $T_l = T$, $\epsilon_e$ becomes constant for all epochs $e$.

# 5 Regret Analysis

After describing UC-CURL algorithm, we now perform the regret and constraint violation analysis. We note that the standard analysis for infinite horizon tabular MDPs of UCRL2 (Jaksch et al., 2010) cannot be directly applied as the policy $\pi_e$ is possibly stochastic for every epoch. Another peculiar aspect of the analysis of the infinite horizon MDPs is that the regret grows linearly with the number of epochs (or policy switches). This is because a new policy induces a new Markov chain and this chain take time to converge to the stationary distribution. The analysis still bounds the regret by $\tilde{O}(T_M S \sqrt{A/T})$ as the number of epochs are bounded by $O(SA \log T)$.

Before diving into the details, we first define few important variables which are key to our analysis. The first variable is the standard $Q$-value function. We define $Q_\gamma^{\pi,P}$ as the long term expected reward on taking action $a$ in state $s$ and then following policy $\pi$ for the MDP with transition $P$. Mathematically, we have

$$Q_\gamma^{\pi,P}(s,a) = r(s,a) + \gamma \sum_{s' \in \mathcal{S}} P(s'|s,a) V_\gamma^{\pi,P}(s'); V_\gamma^{\pi,P}(s) = \mathbb{E}_{a \sim \pi} \left[ Q_\gamma^{\pi,P}(s,a) \right]$$

We also define Bellman error $B^{\pi,\tilde{P}}(s,a)$ for the infinite horizon MDPs as the difference between the cumulative expected rewards obtained for deviating from the system model with transition $\tilde{P}$ for one step by taking action $a$ in state $s$ and then following policy $\pi$. We have:

$$B^{\pi,\tilde{P}}(s,a) = \lim_{\gamma \to 1} \left( Q_\gamma^{\pi,\tilde{P}}(s,a) - r(s,a) - \gamma \sum_{s' \in \mathcal{S}} P(s'|s,a) V_\gamma^{\pi,\tilde{P}}(s,a) \right) \tag{20}$$

After defining the key variables, we can now jump into bounding the objective regret $R(T)$. Intuitively, the algorithm incurs regret on three accounts. First source is following the conservative policy which we require to limit the constraint violations. Second source of regret is solving for the policy which is optimal for the optimistic MDP. Third source of regret is the stochastic behavior of the system. We also note that the constraints are violated because of the imperfect MDP knowledge and the stochastic behavior. However, the conservative behavior actually allows us to violate the constraints within some limits which we will discuss in the later part of this section.

We start by stating our first lemma which bounds the regret due to solving for a conservative policy. We define $\epsilon_e$-tight optimization problem as optimization problem for the true MDP with transitions $P$ with $\epsilon = \epsilon_e$. We bound the gap between the value of function $f$ at the long-term expected reward of the policy for $\epsilon_e$-tight optimization problem and the true optimization problem (Equation (9)-(12)) in the following lemma.

**Lemma 5.1.** *Let $\lambda_{\pi^*}^P$ be the long-term average reward following the optimal feasible policy $\pi^*$ for the true MDP $\mathcal{M}$ and let $\lambda_{\pi_e}^P$ be the long-term average rewards following the optimal policy $\pi_e$ for the $\epsilon_e$ tight optimization problem for the true MDP $\mathcal{M}$, then for $\epsilon_e \leq \delta$, we have,*

$$f\left(\lambda_{\pi^*}^P\right) - f\left(\lambda_{\pi_e}^P\right) \leq 2L\epsilon_e/\delta \tag{21}$$

*Proof Sketch.* We construct a policy for which the steady state distribution is the weighted average of two steady state distributions. First distribution is for the optimal policy for the true optimization problem. Second distribution is for the policy which satisfies Assumption 3.7. We show that this constructed policy satisfies the $\epsilon_e$-tight constraints. Further, using Lipschitz continuity, we convert the difference between function values into the difference between the long-term average rewards to obtain the required result. The detailed proof is provided in Appendix C. $\qquad \square$

Lemma 5.1 and our construction of $\epsilon_e$ sequence allows us to limit the growth of regret because of conservative policy by $\tilde{O}(LdT_M S\sqrt{A/T})$.

To bound the regret from the second source, we use a Bellman error based analysis. In our next lemma, we show that the difference between the performance of a policy on two different MDPs is bounded by long-term averaged Bellman error. Formally, we have:

**Lemma 5.2.** *The difference of long-term average rewards for running the optimistic policy $\pi_e$ on the optimistic MDP, $\lambda_{\pi_e}^{\tilde{P}_e}$, and the average long-term average rewards for running the optimistic policy $\pi_e$ on the*

true MDP, $\lambda_{\pi_e}^P$, is the long-term average Bellman error as

$$\lambda_{\pi_e}^{\tilde{P}_e} - \lambda_{\pi_e}^P = \sum_{s,a} \rho_{\pi_e}^P B^{\pi_e, \tilde{P}_e}(s,a) \tag{22}$$

*Proof Sketch.* We start by writing $Q_\gamma^{\pi_e, \tilde{P}_e}$ in terms of the Bellman error. Subtracting $V_\gamma^{\pi_e, P}$ from $V_\gamma^{\pi_e, \tilde{P}_e}$ and using the fact that $\lambda_{\pi_e}^P = \lim_{\gamma \to 1} V_\gamma^{\pi, P}$ and $\lambda_{\pi_e}^{\tilde{P}_e} = \lim_{\gamma \to 1} V_\gamma^{\pi, \tilde{P}_e}$, we obtain the required result. A complete proof is provided in Appendix D.3. □

After relating the gap between the long-term average rewards of policy $\pi_e$ on the two MDPs, we aim to bound the sum of Bellman error over an epoch. For this, we first bound the Bellman error for a particular state action pair $(s,a)$ in the following lemma. We have,

**Lemma 5.3.** *With probability at least $1 - 1/t_e^6$, the Bellman error $B^{\pi_e, \tilde{P}_e}(s,a)$ for state-action pair $(s,a)$ in epoch $e$ is upper bounded as*

$$B^{\pi_e, \tilde{P}_e}(s,a) \leq \sqrt{\frac{14S \log(2AT)}{1 \vee N_e(s,a)}} \|\tilde{h}\|_\infty \tag{23}$$

*where $N_e(s,a)$ is the number of visitations to $(s,a)$ till epoch $e$ and $\tilde{h}$ is the bias of the MDP with transition probability $\tilde{P}_e$.*

*Proof Sketch.* We start by noting that the Bellman error essentially bounds the impact of the difference in value obtained because of the difference in transition probability to the immediate next state. We bound the difference in transition probability between the optimistic MDP and the true MDP using the result from (Weissman et al., 2003). This approach gives the required result. A complete proof is provided in Appendix D.3. □

We use Lemma 5.2 and Lemma 5.3 to bound the regret because of the imperfect knowledge of the system model. We bound the expected Bellman error in epoch $e$ starting from state $s_{t_e}$ and action $a_{t_e}$ by constructing a Martingale sequence with filtration $\mathcal{F}_t = \{s_1, a_1, \cdots, s_{t-1}, a_{t-1}\}$ and using Azuma's inequality (Bercu et al., 2015). Using the Azuma's inequality, we can also bound the deviations because of the stochasticity of the Markov Decision Process. The result is stated in the following lemma with proof in Appendix D.

**Lemma 5.4.** *With probability at least $1 - T^{-5/4}$, the regret incurred from imperfect model knowledge and process stochastics is bounded by*

$$O(T_M S \sqrt{A(\log AT)/T} + (C T_M S^2 A \log T)/(1 - \rho)) \tag{24}$$

The regret analysis framework also prepares us to bound the constraint violations as well. We again start by quantifying the reasons for constraint violations. The agent violates the constraint because **1.** it is playing with the imperfect knowledge of the MDP and **2.** the stochasticity of the MDP which results in the deviation from the average costs. We note that the conservative policy $\pi_e$ for every epoch does not violate the constraints, but instead allows the agent to manage the constraint violations because of the imperfect model knowledge and the system dynamics.

We note that the Lipschitz continuity of the constraint function $g$ allows us to convert the function of $d$ averaged costs to the sum of $d$ averaged costs. Further, we note that we can treat the cost similar to rewards (Brantley et al., 2020). This property allows us to bound the cost incurred incurred in a way similar to how we bound the gap from the optimal reward by $L d T_M S \sqrt{A(\log AT)/T}$. We now want that the slackness provided by the conservative policy should allow $L d T_M S \sqrt{A(\log AT)/T}$ constraint violations. This is ensured by our chosen $\epsilon_e$ sequence. We formally state that result in the following lemma proven in parts in Appendix D and Appendix E.

**Lemma 5.5.** *The cumulative sum of the $\epsilon_e$ sequence is upper and lower bounded as*

$$\sum_{e=1}^E (t_{e+1} - t_e)\epsilon_e = \Theta\left(K\sqrt{T \log T}\right) \tag{25}$$

After giving the details on bounds on the possible sources of regret and constraint violations, we can formally state the result in the form of following theorem.

**Theorem 5.6.** *For all $T$ and $K = \Theta(LdT_MS\sqrt{A} + CSA/(1-\rho))$, the regret $R(T)$ of UC-CURL algorithm is bounded as*

$$R(T) = O\left(\frac{1}{\delta}LdT_MS\sqrt{A\frac{\log AT}{T}} + \frac{CT_MS^2A\log T}{(1-\rho)T}\right) \tag{26}$$

*and the constraints are bounded as $C(T) = 0$, with probability at least $1 - \frac{1}{T^{5/4}}$.*

### 5.1 Posterior Sampling Algorithm

We can also modify the analysis to obtain Bayesian regret for a posterior sampling version of the UC-CURL algorithm using Lemma 1 of (Osband et al., 2013). In the posterior sampling algorithm, instead of finding the optimistic MDP, we sample the transition probability $\tilde{P}_e$ using an updated posterior. This sampling allows to reduce the complexity of the optimization problem by eliminating Eq. (17) and Eq. (18). The complete algorithm is described in Appendix G. We note that optimization problem for the UC-CURL algorithm is feasible because the true MDP lies in the confidence interval. However, for the sampled MDP obtaining the feasibility requires a stronger Slater's condition.

### 5.2 Further Modifications

The proposed algorithm, and the analysis can be easily extended to $M$ convex constraints $g_1, \cdots, g_M$ by applying union bounds. Further, our analysis uses Proposition 1 of (Jaksch et al., 2010) to bound the epochs by $O(SA\log_2 T)$. However, we can improve the empirical performance of the UC-CURL algorithm by modifying the epoch trigger condition (Line 11 of Algorithm 1). Triggering a new episode whenever $\nu_e(s,a)$ becomes $\max\{1, \nu_{e-1}(s,a)+1\}$ for any state-action pair results in a linearly increasing episode length with total epochs bounded by $O(SA + \sqrt{SAT})$. This modification results in a better empirical performance (See Appendix 6 for simulations) at the cost of a higher theoretical regret bound and computation complexity for obtaining a new policy at every epoch.

## 6 Simulation Results

To validate the performance of the UC-CURL algorithm and the PS-CURL algorithm, we run the simulation on the flow and service control in a single-serve queue, which was introduced in (Altman & Schwartz, 1991). Along with validating the performance of the proposed algorithms, we also compare the algorithms against the algorithms proposed in (Singh et al., 2020) and in (Chen et al., 2022) for model-based constrained reinforcement learning for infinite horizon MDPs. Compared to these algorithms, we note that our algorithm is also designed to handle concave objectives of expected rewards with convex constraints on costs with 0 constraint violations.

In the queue environment, a discrete-time single-server queue with a buffer of finite size $L$ is considered. The number of customers waiting in the queue is considered as the state in this problem and thus $|S| = L+1$. Two kinds of the actions, service and flow, are considered in the problem and control the number of customers together. The action space for service is a finite subset $A$ in $[a_{min}, a_{max}]$, where $0 < a_{min} \le a_{max} < 1$. Given a specific service action $a$, the service a customer is successfully finished with the probability $b$. If the service is successful, the length of the queue will reduce by 1. Similarly, the space for flow is also a finite subsection $B$ in $[b_{min}, b_{max}]$. In contrast to the service action, flow action will increase the queue by 1 with probability $b$ if the specific flow action $b$ is given. Also, we assume that there is no customer arriving when the queue is full. The overall action space is the Cartesian product of the $A$ and $B$. According to the service and flow probability, the transition probability can be computed and is given in the Table 2.

Define the reward function as $r(s,a,b)$ and the constraints for service and flow as $c^1(s,a,b)$ and $c^2(s,a,b)$, respectively. Define the stationary policy for service and flow as $\pi_a$ and $\pi_b$, respectively. Then, the problem

Table 2: Transition probability of the queue system

| Current State | $P(x_{t+1} = x_t - 1)$ | $P(x_{t+1} = x_t)$ | $P(x_{t+1} = x_t + 1)$ |
|---|---|---|---|
| $1 \leq x_t \leq L - 1$ | $a(1-b)$ | $ab + (1-a)(1-b)$ | $(1-a)b$ |
| $x_t = L$ | $a$ | $1-a$ | $0$ |
| $x_t = 0$ | $0$ | $1 - b(1-a)$ | $b(1-a)$ |

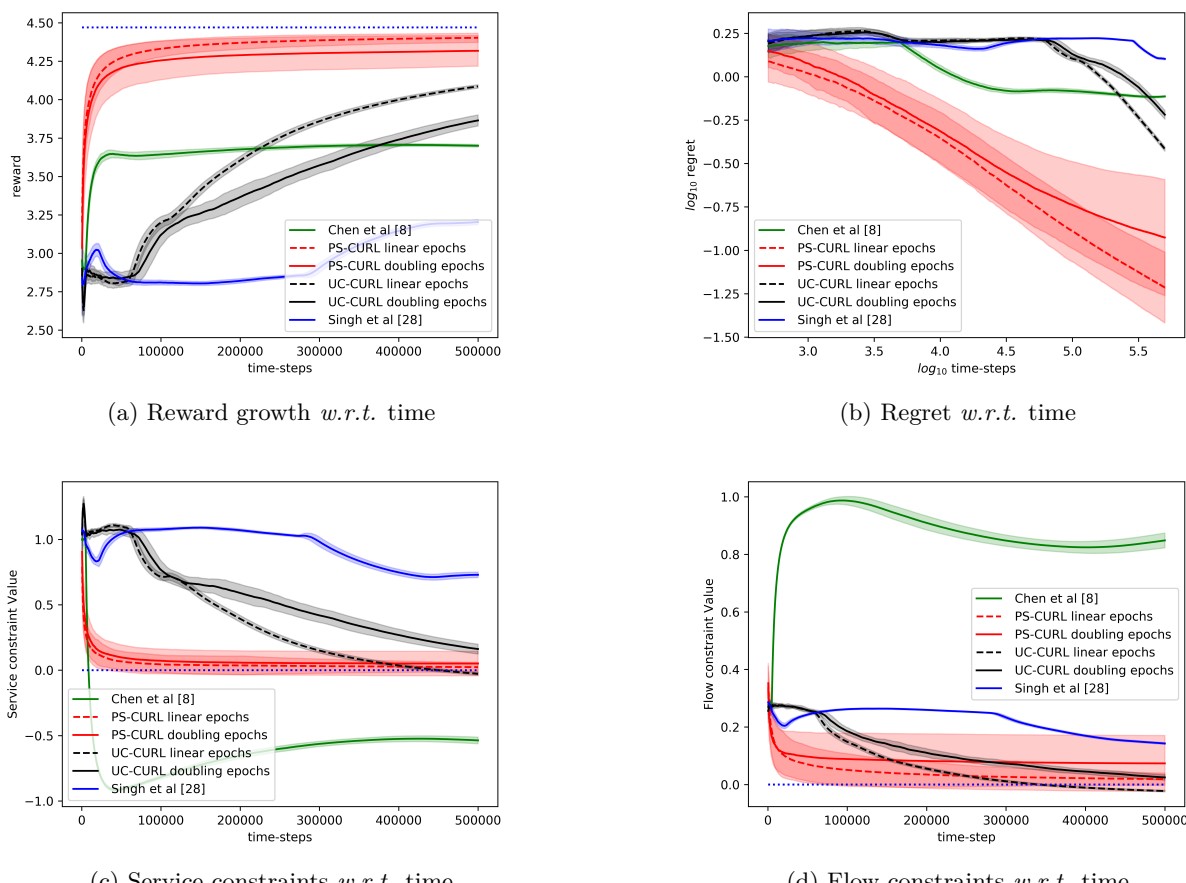

(a) Reward growth *w.r.t.* time

(b) Regret *w.r.t.* time

(c) Service constraints *w.r.t.* time

(d) Flow constraints *w.r.t.* time

Figure 1: Performance of the proposed UC-CURL and PS-CURL algorithms on a flow and service control problem for a single queue with doubling epoch lengths and linearly increasing epoch lengths. The algorithms are compared against Chen et al. (2022) and Singh et al. (2020)

can be defined as                                                   .

$$\max_{\pi_a, \pi_b} \quad \lim_{T \to \infty} \frac{1}{T} \sum_{t=1}^{T} r(s_t, \pi_a(s_t), \pi_b(s_t))$$

$$s.t. \quad \lim_{T \to \infty} \frac{1}{T} \sum_{t=1}^{T} c^1(s_t, \pi_a(s_t), \pi_b(s_t)) \geq 0 \tag{27}$$

$$\lim_{T \to \infty} \frac{1}{T} \sum_{t=1}^{T} c^2(s_t, \pi_a(s_t), \pi_b(s_t)) \geq 0$$

According to the discussion in (Altman & Schwartz, 1991), we define the reward function as $r(s, a, b) = 5 - s$, which is an decreasing function only dependent on the state. It is reasonable to give higher reward when the

number of customers waiting in the queue is small. For the constraint function, we define $c^1(s, a, b) = -10a + 6$ and $c^2 = -8(1 - b)^2 + 2$, which are dependent only on service and flow action, respectively. Higher constraint value is given if the probability for the service and flow are low and high, respectively.

In the simulation, the length of the buffer is set as $L = 5$. The service action space is set as $[0.2, 0.4, 0.6, 0.8]$ and the flow action space is set as $[0.4, 0.5, 0.6, 0.7]$. We use the length of horizon $T = 5 \times 10^5$ and run 50 independent simulations of all algorithms. The experiments were run on a 36 core Intel-i9 CPU @3.00 GHz with 64 GB of RAM. The result is shown in the Figure 1. The average values of the cumulative reward and the constraint functions are shown in the solid lines. Further, we plot the standard deviation around the mean value in the shadow to show the random error. In order to compare this result to the optimal, we assume that the full information of the transition dynamics is known and then use Linear Programming to solve the problem. The optimal cumulative reward for the constrained optimization is calculated to be 4.48 with both flow constraint and service constraint values to be 0. The optimal cumulative reward for the unconstrained optimization is 4.8 with service constraint being $-2$ and flow constraint being $-0.88$.

We now discuss the performance of all the algorithms starting with our algorithms UC-CURL and PS-CURL. In Figure 1, we observe that the proposed UC-CURL algorithm in Algorithm 1 does not perform well initially. We observe that this is because the confidence interval radius $\sqrt{S \log(At)/N(s, a)}$ for any $(s, a)$ are not tight enough in the initial period. After the algorithms collects sufficient samples to construct tight confidence intervals around the transition probabilities, the algorithm starts converging towards the optimal policy. We also note that the linear epoch modification of the algorithm works better than the doubling epoch algorithm presented in Algorithm 1. This is because the linear epoch variant updates the policy quickly whereas the doubling epoch algorithm works with the same policy for too long and thus looses the advantages of collected samples. For our implementation, we choose the value of parameter $K$ in Algorithm 1 as $K = 1$, using which we observe that the constraint values start converging towards zero.

We now analyse the performance of the PS-CURL algorithm. For our implementation of the PS-CURL algorithm, we sample the transition probabilities using Dirichlet distribution. Note that the true transition probabilities were not sampled from a Dirichlet distribution and hence this experiment also shows the robustness against misspecified priors. We observe that the algorithm quickly brings the reward close to the optimal rewards. The performance of the PS-CURL algorithm is significantly better than the UC-CURL algorithm. We suspect this is because the UC-CURL algorithm wastes a large-number of steps to find optimistic policy with a large confidence interval. This observation aligns with the TDSE algorithm (Ouyang et al., 2017), where they show that the Thompson sampling algorithm with $O(\sqrt{SAT})$ epochs performs empirically better than the optimism based UCRL2 algorithm (Jaksch et al., 2010) with $O(\sqrt{SA \log T})$ epochs. (Osband et al., 2013) also made a similar observation where their PSRL algorithm worked better than the UCRL2 algorithm. Again, we set the value of parameter $K$ as 1 and with $K = 1$, the algorithm does not violate constraints. We also observe that the standard deviation of the rewards and constraints are higher for the PS-CURL algorithm as compared to the UC-CURL algorithm as the PS-CURL algorithm has an additional stochastic component which arises from sampling the transition probabilities.

After analysing the algorithms presented in this paper, we now analyse the performance of the algorithm by Chen et al. (2022). They provide an optimistic online mirror descent algorithm which also works with conservative parameter to tightly bound constraint violations. Their algorithm also obtains a $O(\sqrt{T})$ regret bound. However, their algorithm is designed for a linear reward/constraint setup with a single constraint, and empirically the algorithm is difficult to tune as it requires additional knowledge of $T_M$, $\rho$, $\delta$, and $T$ to fine tune parameters used in their algorithm. We set the value of the learning rate $\theta$ for online mirror descent as $5 \times 10^{-2}$ with an episode length of $5 \times 10^3$. Further, we scale the rewards and costs to ensure that they lie between 0 and 1. We analyze the behavior of the optimistic online mirror descent algorithm in Figure 1b. We observe that the algorithm has three phases. The first phase is the first episodes where the algorithm uses a uniform policy which is the initial flat area till first 5000 steps. In the second phase, the algorithm updates the policy for the first time and starts converging to the optimal policy with a convergence rate which matches to that of the PS-CURL algorithm. However, after few policy updates, we observe that the algorithm has oscillatory behavior which is because the dual variable updates require online constraint violations.

Finally, we analyze the the algorithm by Singh et al. (2020). They also provide an algorithm which proceeds in epochs and solves an optimization problem at every epoch. The algorithm considers a fixed epoch length $T^{1/3}$. Further, the algorithm considers a confidence interval on each estimate of $P(s'|s, a)$ for all $s, a, s'$

triplet. The algorithm does not perform well even though it updates the policy most frequently because of creating confidence intervals on individual transition probabilities $P(s'|s, a)$ instead of the probability vector $P(s'|s, a)$.

From the experimental observations, we note that the proposed UC-CURL algorithm is suitable in cases where the parameter tuning is not possible and the system requires tighter bounds on deviation of the performance of the algorithm. The PS-CURL algorithm can be used in cases where the variance in algorithm's performance can be tolerated or computational complexity is a constraint. Further, for both the algorithms, it is beneficial to use the linear increasing epoch lengths. Additionally, the algorithm by Chen et al. (2022) is suitable for cases where solving an optimization equation is not feasible, for example an embedded system, as the algorithm updates policy using exponential function which can be easily computed. However, this algorithm is only applicable in applications with linear reward/constraint and single constraint.

## 7  Conclusion

We considered the problem of Markov Decision Process with concave objective and convex constraints. For this problem, we proposed UC-CURL algorithm which works on the principle of optimism. To bound the constraint violations, we solve for a conservative policy using an optimistic model for an $\epsilon$-tight optimization problem. Using an analysis based on Bellman error for infinite-horizon MDPs, we show the UC-CURL algorithm achieves 0 constraint violations with a regret bound of $\tilde{O}(LdT_M S\sqrt{A/T} + (CSA\log T)/(T(1-\rho)))$. Further, to reduce the computation complexity of finding optimistic MDP, we also propose a posterior sampling algorithm which finds the optimal policy for a sampled MDP. We provide a Bayesian regret bound of $\tilde{O}(LdT_M S\sqrt{A/T} + (CT_M S^2 A\log T)/(T(1-\rho)))$ for the posterior sampling algorithm by considering a stronger Slater's condition to solve for constrained optimization for sampled MDPs as well. As part of potential future works, we consider dynamically configuring $K$ to be an interesting and important direction to reduce the requirement of problem parameters.

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

# A    Assumptions and their justification

We first introduce our initial assumptions on the MDP $\mathcal{M}$. We assume the MDP $\mathcal{M}$ is ergodic. Ergodicity is a commonly used assumption in constrained RL literature (Singh et al., 2020; Chen et al., 2022). Further, ergodicity is required to obtain stationary Markovian policies which can be tranferred from training setup to test environment. Let $P_{\pi,s}^{t}$ denote the $t$-step transition probability on following policy $\pi$ in MDP $\mathcal{M}$ starting from some state $s$. Also, let $T_{s \to s'}^{\pi}$ denotes the time taken by the Markov chain induced by the policy $\pi$ to hit state $s'$ starting from state $s$. Building on these variables, $P_{\pi,s}^{t}$ and $T_{s \to s'}^{\pi}$, we make our first assumption as follows:

**Assumption A.1.** The MDP $\mathcal{M}$ is ergodic, or

$$\|P_{\pi,s}^{t} - P_{\pi}\| \leq C\rho^{t} \tag{28}$$

where $P_{\pi}$ is the long-term steady state distribution induced by policy $\pi$, and $C > 0$ and $\rho < 1$ are problem specific constants. Also, we have

$$T_M := \max_{\pi} \mathbb{E}[T_{s \to s'}^{\pi}] < \infty \tag{29}$$

where $T_M$ is the finite mixing time of the MDP $\mathcal{M}$.

We note that in most of the problems, rewards are engineered according to the problem. However, the system dynamics are stochastic and typically not known. Based on this, we make the following assumption on rewards.

**Assumption A.2.** The rewards $r(s,a)$, the costs $c_i(s,a); \forall\, i$ and the functions $f$ and $g$ are known to the agent.

Our next assumption is on the functions $f$ and $g$. Many practically implemented fairness objectives are concave (Kwan et al., 2009), or the agent want to explore all possible state action pairs by maximizing the entropy of the long-term state-action distribution (Hazan et al., 2019), or the agent may want to minimize divergence with respect to a certain expert policy (Ghasemipour et al., 2020). Formally, we have

**Assumption A.3.** The scalarization function $f$ is jointly concave and the constraints $g$ are jointly convex. Hence for any arbitrary distributions $\mathcal{D}_1$ and $\mathcal{D}_2$, the following holds.

$$f\left(\mathbb{E}_{x \sim \mathcal{D}_1}\left[x\right]\right) \geq \mathbb{E}_{x \sim \mathcal{D}_1}\left[f\left(x\right)\right] \tag{30}$$

$$g\left(\mathbb{E}_{\mathbf{x} \sim \mathcal{D}_2}\left[\mathbf{x}\right]\right) \leq \mathbb{E}_{\mathbf{x} \sim \mathcal{D}_2}\left[g\left(\mathbf{x}\right)\right];\ \mathbf{x} \in \mathbb{R}^d \tag{31}$$

We impose an additional assumption on the functions $f$ and $g$. We assume that the functions are continuous and Lipschitz continuity in particular. Lipschitz continuity is a common assumption for optimization literature (Bubeck et al., 2015; Jin et al., 2017; Zhang et al., 2020). Additionally, in practice this assumption is validated, often by adding some regularization. We have,

**Assumption A.4.** The function $f$ and $g$ are assumed to be a $L-$ Lipschitz function, or

$$|f\left(x\right) - f\left(y\right)| \leq L|x - y|;\ x, y \in \mathbb{R} \tag{32}$$

$$|g\left(\mathbf{x}\right) - g\left(\mathbf{y}\right)| \leq L\left\|\mathbf{x} - \mathbf{y}\right\|_1;\ \mathbf{x}, \mathbf{y} \in \mathbb{R}^d \tag{33}$$

We consider a standard setup of concave and the Lipschitz function as considered by (Cheung, 2019; Brantley et al., 2020; Yu et al., 2021). Note that the analysis in this paper directly works for $f: \mathbb{R}^K \to \mathbb{R}$, where the function takes as input multiple average per-step rewards. We can obtain maximum entropy exploration if choose function $f = -\sum_k \lambda_k \log(\lambda_k + \eta)$ with $r_k(s,a) = \mathbf{1}_{\{s_k,a_k\}}$ for a particular state action pair $s_k, a_k$ and choosing $K = S \times A$ to cover all state-action pairs and a regularizer $\eta$.

Next, we assume the following Slater's condition to hold.

**Assumption A.5.** There exists a policy $\pi$, and one constant $\delta > LdST_M\sqrt{A/T}$ such that

$$g\left(\zeta_{\pi}^P(1), \cdots, \zeta_{\pi}^P(d)\right) \leq -\delta \tag{34}$$

Further, if there is a (possibly unknown) lower bound on time-horizon, $T_l \geq \exp(1)$, then we only require $\delta > LdST_M\sqrt{A(\log T_l)/T_l}$. This assumption is again a standard assumption in the constrained RL literature (Efroni et al., 2020; Ding et al., 2021; 2020; Wei et al., 2022a). $\delta$ is referred as Slater's constant. (Ding et al., 2021) assumes that the Slater's constant $\delta$ is known. (Wei et al., 2022a) assumes that the number of iterations of the algorithm is at least $\tilde{\Omega}(SAH/\delta)^5$ for episode length $H$. On the contrary, we simply assume the existence of $\delta$ and a lower bound on the value of $\delta$ which can be relaxed as the agent acquires more time to interact with the environment.

## B  Efficiently solving the Conservative Optimistic Optimization problem

We now provide the details on efficiently solving the optimistic optimization problem described with constraints in Equation (16)-(18). Similar to the method proposed in (Rosenberg & Mansour, 2019), we define a new variable $p(s, a, s')$ which denotes the probability of being is state $s$, taking action $a$, and then moving to state $s'$. Now, the transition probability to next state $s'$ given current state $s$ and action $a$ is given as:

$$P(s'|s,a) = \frac{p(s,a,s')}{\sum_{s'} p(s,a,s')} \tag{35}$$

Further, the occupancy measure of state-action pair $s, a$ is given as

$$\rho(s,a) = \sum_{s'} p(s,a,s') \tag{36}$$

Based on these two observations, at the beginning of epoch $e$, we define the optimization problem as follows:

$$\max_{p(s,a,s')} f\left(\sum_{s,a}\left(\left(\sum_{s'}p(s,a,s')\right)r(s,a)\right)\right) \tag{37}$$

subject to following constraints

$$\sum_{s,a,s'} p(s,a,s') = 1, p(s,a,s') \geq 0 \tag{38}$$

$$\sum_{s',a} p(s,a,s') = \sum_{s',a} p(s',a,s) \tag{39}$$

$$g\left(\sum_{s,a}\left(\left(\sum_{s'}p(s,a,s')\right)c_1(s,a)\right), \cdots, \sum_{s,a}\left(\left(\sum_{s'}p(s,a,s')\right)c_d(s,a)\right)\right) \leq \epsilon_e \tag{40}$$

$$p(s,a,s') - \frac{\hat{P}(s,a,s')}{1 \vee N_e(s,a)}\sum_{s'}p(s,a,s') \leq \alpha(s,a,s') \tag{41}$$

$$\frac{\hat{P}(s,a,s')}{1 \vee N_e(s,a)}\sum_{s'}p(s,a,s') - p(s,a,s') \leq \alpha(s,a,s') \tag{42}$$

$$\sum_{s'}\alpha(s,a,s') \leq \sqrt{\frac{14S\log(2At)}{1 \vee N_e(s,a)}}\sum_{s'}p(s,a,s') \tag{43}$$

for all $s \in \mathcal{S}, a \in \mathcal{A}$, and $s' \in \mathcal{S}$. Also, $\alpha(s,a,s')$ is an auxiliary variable introduced to reduce the complexity of $\ell_1$ norm constraints and the present the optimization problem in a disciplined convex program which can be coded easily in CVXPY. The Equations (41), (42), (43) jointly describe the $\ell_1$ confidence interval on the probability estimates.

## C   Proof of Lemma 5.1

*Proof.* Note that $\rho_{\pi^*}^P$ denotes the stationary distribution of the optimal solution which satisfies

$$g\left(\sum_{s,a}\rho_{\pi^*}^P(s,a)c_1(s,a),\cdots,\sum_{s,a}\rho_{\pi^*}^P(s,a)c_d(s,a)\right)\le C \tag{44}$$

Further, from Assumption 3.7, we have a feasible policy $\pi$ for which

$$g\left(\sum_{s,a}\rho_{\pi}^P(s,a)c_1(s,a),\cdots,\sum_{s,a}\rho_{\pi}^P(s,a)c_d(s,a)\right)\le C-\delta \tag{45}$$

We now construct a stationary distribution $\rho^P$ obtain the corresponding $\pi'_e$ as:

$$\rho^P(s,a)=\left(1-\frac{\epsilon_e}{\delta}\right)\rho_{\pi^*}^P(s,a)+\frac{\epsilon_e}{\delta}\rho_{\pi}^P(s,a) \tag{46}$$

$$\pi'_e=\rho^P(s,a)/\left(\sum_{s,b}\rho^P(s,b)\right) \tag{47}$$

For this new policy and convex constraint $g$, we observe that

$$g\left(\sum_{s,a}\rho_{\pi}^P(s,a)c_1(s,a),\cdots,\sum_{s,a}\rho_{\pi}^P(s,a)c_d(s,a)\right) \tag{48}$$

$$=g\left(\sum_{s,a}\left(\left(1-\frac{\epsilon_e}{\delta}\right)\rho_{\pi^*}^P+\frac{\epsilon_e}{\delta}\rho_{\pi}^P\right)(s,a)c_1(s,a),\cdots,\sum_{s,a}\left(\left(1-\frac{\epsilon_e}{\delta}\right)\rho_{\pi^*}^P+\frac{\epsilon_e}{\delta}\rho_{\pi}^P\right)(s,a)c_d(s,a)\right) \tag{49}$$

$$\le\left(1-\frac{\epsilon_e}{\delta}\right)g\left(\sum_{s,a}\rho_{\pi^*}^P(s,a)c_1(s,a),\cdots,\sum_{s,a}\rho_{\pi^*}^P(s,a)c_d(s,a)\right)$$

$$+\frac{\epsilon_e}{\delta}g\left(\sum_{s,a}\rho_{\pi}^P(s,a)c_1(s,a),\cdots,\sum_{s,a}\rho_{\pi}^P(s,a)c_d(s,a)\right) \tag{50}$$

$$\le\left(1-\frac{\epsilon_e}{\delta}\right)C+\frac{\epsilon_e}{\delta}\left(C-\delta\right) \tag{51}$$

$$=C-\delta\le C-\epsilon_e \tag{52}$$

where Equation (50) follows from the convexity of the constraints. Equation (51) follows from Equation (44) and Equation (45).

Note that the policy $\pi'_e$ corresponding to stationary distribution constructed in Equation (46) satisfies the $\epsilon_e$-tight constraints. Further, we find $\pi_e^*$ as the optimal solution for the $\epsilon_e$-tight optimization problem. Hence, we have

$$f\left(\sum_{s,a}\rho_{\pi^*}^P(s,a)r(s,a)\right)-f\left(\sum_{s,a}\rho_{\pi_e^*}^P(s,a)r(s,a)\right)$$

$$\le f\left(\sum_{s,a}\rho_{\pi^*}^P(s,a)r(s,a)\right)-f\left(\sum_{s,a}\rho^P(s,a)r(s,a)\right) \tag{53}$$

$$\le L\left|\sum_{s,a}\left(\rho_{\pi^*}^P(s,a)-\rho^P(s,a)\right)r(s,a)\right| \tag{54}$$

$$\le L\left|\sum_{s,a}\left(\rho_{\pi^*}^P(s,a)-\left(1-\frac{\epsilon_e}{\delta}\right)\rho_{\pi^*}^P(s,a)-\frac{\epsilon_e}{\delta}\rho_{\pi}^P(s,a)\right)c_d(s,a)\right| \tag{55}$$

$$\leq \quad L\frac{\epsilon_e}{\delta}\Big|\sum_{s,a}\left(\rho_{\pi^*}^P(s,a)-\rho_\pi^P(s,a)\right)r(s,a)\Big| \tag{56}$$

$$\leq \quad L\frac{\epsilon_e}{\delta}\Big|\sum_{s,a}\rho_{\pi^*}^P r(s,a)\Big|+L\frac{\epsilon_e}{\delta}\Big|\sum_{s,a}\rho_\pi^P(s,a)r(s,a)\Big| \tag{57}$$

$$\leq \quad 2L\frac{\epsilon_e}{\delta} \tag{58}$$

where, Equation (54) follows from the Lipschitz assumption on the joint objective $f$. Equation (58) follows from the fact that $r(s,a)\leq 1$ for all $(s,a)\in\mathcal{S}\times\mathcal{A}$. $\qquad\square$

## D  Objective Regret Bound

In this section, we begin with breaking down the regret in multiple components and then analysis the components individually.

### D.1  Regret breakdown

We first break down our regret into multiple parts which will help us bound the regret.

$$R(T)=f(\lambda_*^P)-f\left(\frac{1}{T}\sum_{t=1}^T r_t(s_t,a_t)\right) \tag{59}$$

$$=f(\lambda_*^P)-\frac{1}{T}\sum_{e=1}^E T_e f(\lambda_{\pi_e^*}^P)+\frac{1}{T}\sum_{e=1}^E T_e f(\lambda_{\pi_e^*}^P)-f\left(\frac{1}{T}\sum_{t=1}^T r_t(s_t,a_t)\right) \tag{60}$$

$$=\frac{1}{T}\sum_{e=1}^E T_e\left(f(\lambda_*^P)-f(\lambda_{\pi_e^*}^P)\right)+\frac{1}{T}\sum_{e=1}^E T_e f(\lambda_{\pi_e^*}^P)-f\left(\frac{1}{T}\sum_{t=1}^T r_t(s_t,a_t)\right) \tag{61}$$

$$\leq\frac{1}{T}\sum_{e=1}^E T_e\left(f(\lambda_*^P)-f(\lambda_{\pi_e^*}^P)\right)+\frac{1}{T}\sum_{e=1}^E T_e f(\lambda_{\pi_e}^{\tilde{P}_e})-f\left(\frac{1}{T}\sum_{t=1}^T r_t(s_t,a_t)\right) \tag{62}$$

$$\leq\frac{1}{T}\sum_{e=1}^E T_e\left(f(\lambda_*^P)-f(\lambda_{\pi_e^*}^P)\right)+f\left(\frac{1}{T}\sum_{e=1}^E T_e\lambda_{\pi_e}^{\tilde{P}_e}\right)-f\left(\frac{1}{T}\sum_{t=1}^T r_t(s_t,a_t)\right) \tag{63}$$

$$\leq\frac{1}{T}\sum_{e=1}^E T_e\left(f(\lambda_*^P)-f(\lambda_{\pi_e^*}^P)\right)+L\Big|\frac{1}{T}\sum_{e=1}^E T_e\lambda_{\pi_e}^{\tilde{P}_e}-\frac{1}{T}\sum_{t=1}^T r_t(s_t,a_t)\Big| \tag{64}$$

$$=\frac{1}{T}\sum_{e=1}^E T_e\left(f(\lambda_*^P)-f(\lambda_{\pi_e^*}^P)\right)+L\Big|\frac{1}{T}\sum_{e=1}^E\sum_{t=t_e}^{t_{e+1}-1}\left(\lambda_{\pi_e}^{\tilde{P}_e}-\lambda_{\pi_e}^P+\lambda_{\pi_e}^P-r_t(s_t,a_t)\right)\Big| \tag{65}$$

$$\leq\frac{1}{T}\sum_{e=1}^E T_e\left(f(\lambda_*^P)-f(\lambda_{\pi_e^*}^P)\right)+L\Big|\frac{1}{T}\sum_{e=1}^E\sum_{t=t_e}^{t_{e+1}-1}\left(\lambda_{\pi_e}^{\tilde{P}_e}-\lambda_{\pi_e}^P\right)\Big|$$

$$+L\Big|\frac{1}{T}\sum_{e=1}^E\sum_{t=t_e}^{t_{e+1}-1}\left(\lambda_{\pi_e}^P-r_t(s_t,a_t)\right)\Big| \tag{66}$$

$$=R_1(T)+R_2(T)+R_3(T) \tag{67}$$

where Equation (62) comes from the fact that the policy $\pi_e$ is for the optimistic CMDP and provides a higher value of the function $f$. Equation 63 comes from the concavity of the function $f$, and Equation 64 comes from the Lipschitz continuity of the function $f$. The three terms in Equation (67) are now defined as:

$$R_1(T)=\frac{L}{T}\sum_{e=1}^E T_e\left(f(\lambda_*^P)-f(\lambda_{\pi_e^*}^P)\right) \tag{68}$$

$R_1(T)$ denotes the regret incurred from not playing the optimal policy $\pi^*$ for the true optimization problem in Equation (9) but the optimal policy $\pi_e^*$ for the $\epsilon_e$-tight optimization problem in epoch $e$.

$$R_2(T) = \frac{L}{T} \Big| \sum_{e=1}^{E} \sum_{t=t_e}^{t_{e+1}-1} \left( \lambda_{\pi_e}^{\tilde{P}_e} - \lambda_{\pi_e}^{P} \right) \Big| \tag{69}$$

$R_2(T)$ denotes the gap between expected rewards from playing the optimal policy $\pi_e$ for $\epsilon_e$-tight optimization problem on the optimistic MDP instead of the true MDP. For this term, we further consider another modification. We have $\tilde{P}_e$ being the optimistic MDP with optimistic policy $\pi_e$ as the solutions for the optimization equation solved at the beginning of every epoch. Now consider an MDP $\tilde{P}_e^*$ in the confidence set, which maximizes the long term expected reward for policy $\pi_e$ or $\lambda_{\pi_e}^{\tilde{P}_e^*} \geq \lambda_{\pi_e}^{P_e}$ for all $P_e$ in the confidence interval at epoch $e$. Hence, we have

$$R_2(T) = \frac{L}{T} \Big| \sum_{e=1}^{E} \sum_{t=t_e}^{t_{e+1}-1} \left( \lambda_{\pi_e}^{\tilde{P}_e} - \lambda_{\pi_e}^{P} \right) \Big| \tag{70}$$

$$\leq \frac{L}{T} \Big| \sum_{e=1}^{E} \sum_{t=t_e}^{t_{e+1}-1} \left( \lambda_{\pi_e}^{\tilde{P}_e^*} - \lambda_{\pi_e}^{P} \right) \Big| \tag{71}$$

We relabel $\tilde{P}_e^*$ as $\tilde{P}_e$ in the remaining analysis to reduce notation clutter.

$$R_3(T) = \frac{L}{T} \Big| \sum_{e=1}^{E} \sum_{t=t_e}^{t_{e+1}-1} \left( \lambda_{\pi_e}^{P} - r_t(s_t, a_t) \right) \Big| \tag{72}$$

$R_3(T)$ denotes the gap between obtained rewards from playing the optimal policy $\pi_e$ for $\epsilon_e$-tight optimization problem the true MDP and the expected per-step reward of playing the optimal policy $\pi_e$ for $\epsilon_e$-tight optimization problem the true MDP.

## D.2   Bounding $R_1(T)$

Bounding $R_1(T)$ uses Lemma 5.1. We have the following set of equations:

$$R_1(T) = \frac{1}{T} \sum_{e=1}^{E} \sum_{t=t_e}^{t_{e+1}-1} \left( f(\lambda_*^P) - f(\lambda_{\pi_e}^P) \right) \tag{73}$$

$$\leq \frac{1}{T} \sum_{e=1}^{E} \sum_{t=t_e}^{t_{e+1}-1} \frac{2L\epsilon_e}{\delta} \tag{74}$$

$$= \frac{2L}{T\delta} \sum_{e=1}^{E} \sum_{t=t_e}^{t_{e+1}-1} K\sqrt{\frac{\log t}{t}} \tag{75}$$

$$= \frac{2KL}{T\delta} \sum_{t=1}^{T} \sqrt{\frac{\log t}{t}} \tag{76}$$

$$\leq \frac{2KL}{T\delta} \sum_{t=1}^{T} \sqrt{\frac{\log T}{t}} \tag{77}$$

$$= \frac{2KL \log T}{T\delta} (1 + \sum_{t=2}^{T} \sqrt{\frac{1}{t}}) \tag{78}$$

$$\leq \frac{2KL \log T}{T\delta} (1 + \int_{t=1}^{T} \sqrt{\frac{1}{t}} dt) \tag{79}$$

$$\leq \frac{2KL \log T}{T\delta} (2\sqrt{T}) \tag{80}$$

where Equation (77) follows from the fact that $\log t \leq \log T$ for all $t \leq T$.

## D.3   Bounding $R_2(T)$

We relate the difference between long-term average rewards for running the optimistic policy $\pi_e$ on the optimistic MDP $\lambda_{\pi_e}^{\tilde{P}_e}$ and the long-term average rewards for running the optimistic policy $\pi_e$ on the true MDP $(\lambda_{\pi_e}^P)$ with the Bellman error. Formally, we have the following lemma:

**Lemma D.1.** *The difference of long-term average rewards for running the optimistic policy $\pi_e$ on the optimistic MDP, $\lambda_{\pi_e}^{\tilde{P}_e}$, and the average long-term average rewards for running the optimistic policy $\pi_e$ on the true MDP, $\lambda_{\pi_e}^P$, is the long-term average Bellman error as*

$$\lambda_{\pi_e}^{\tilde{P}_e} - \lambda_{\pi_e}^P = \sum_{s,a} \rho_{\pi_e}^P B^{\pi_e, \tilde{P}_e}(s, a) \tag{81}$$

*Proof.* Note that for all $s \in \mathcal{S}$, we have:

$$V_\gamma^{\pi_e, \tilde{P}_e}(s) = \mathbb{E}_{a \sim \pi_e} \left[ Q_\gamma^{\pi_e, \tilde{P}_e}(s, a) \right] \tag{82}$$

$$= \mathbb{E}_{a \sim \pi_e} \left[ B^{\pi_e, \tilde{P}_e}(s, a) + r(s, a) + \gamma \sum_{s' \in \mathcal{S}} P(s'|s, a) V_\gamma^{\pi_e, \tilde{P}_e}(s') \right] \tag{83}$$

where Equation (83) follows from the definition of the Bellman error for state action pair $(s, a)$.

Similarly, for the true MDP, we have,

$$V_\gamma^{\pi_e,P}(s) = \mathbb{E}_{a \sim \pi_e}\left[Q_\gamma^{\pi_e,}(s,a)\right] \tag{84}$$

$$= \mathbb{E}_{a \sim \pi_e}\left[r(s,a) + \gamma \sum_{s' \in \mathcal{S}} P(s'|s,a)V_\gamma^{\pi_e,P}(s')\right] \tag{85}$$

Subtracting Equation (85) from Equation (83), we get:

$$V_\gamma^{\pi_e,\tilde{P}_e}(s) - V_\gamma^{\pi_e,P}(s) = \mathbb{E}_{a \sim \pi_e}\left[B^{\pi_e,\tilde{P}_e}(s,a) + \gamma \sum_{s' \in \mathcal{S}} P(s'|s,a)\left(V_\gamma^{\pi_e,\tilde{P}_e} - V_\gamma^{\pi_e,\tilde{P}_e}\right)(s')\right] \tag{86}$$

$$= \mathbb{E}_{a \sim \pi_e}\left[B^{\pi_e,\tilde{P}_e}(s,a)\right] + \gamma \sum_{s' \in \mathcal{S}} P_{\pi_e}\left(V_\gamma^{\pi_e,\tilde{P}_e} - V_\gamma^{\pi_e,\tilde{P}_e}\right)(s') \tag{87}$$

Using the vector format for the value functions, we have,

$$\bar{V}_\gamma^{\pi_e,\tilde{P}_e} - \bar{V}_\gamma^{\pi_e,P} = (I - \gamma P_{\pi_e})^{-1}\overline{B}_{\pi_e}^{\pi_e,\tilde{P}_e} \tag{88}$$

Now, converting the value function to average per-step reward we have,

$$\lambda_{\pi_e}^{\tilde{P}_e}\mathbf{1}_S - \lambda_{\pi_e}^P\mathbf{1}_S = \lim_{\gamma \to 1}(1-\gamma)\left(\bar{V}_\gamma^{\pi_e,\tilde{P}_e} - \bar{V}_\gamma^{\pi_e,P}\right) \tag{89}$$

$$= \lim_{\gamma \to 1}(1-\gamma)(I - \gamma P_{\pi_e})^{-1}\overline{B}_{\pi_e}^{\pi_e,\tilde{P}_e} \tag{90}$$

$$= \left(\sum_{s,a}\rho_{\pi_e}^P B^{\pi_e,\tilde{P}_e}(s,a)\right)\mathbf{1}_S \tag{91}$$

where the last equation follows from the definition of occupancy measures by (Puterman, 2014). □

*Remark* D.2. Note that the Bellman error is not to be confused by Advantage function and policy improvement lemma (Langford & Kakade, 2002). The policy improvement lemma relates the performance of two policies on same MDP whereas we bounded the performance of one policy on two different MDPs in Lemma D.1

We now want to bound the Bellman errors to bound the gap between the average per-step reward $\lambda_{\pi_e}^{\tilde{P}_e}$, and $\lambda_{\pi_e}^P$. From the definition of Bellman error and the confidence intervals on the estimated transition probabilities, we obtain the following lemma:

**Lemma D.3.** *With probability at least $1 - 1/t_e^6$, the Bellman error $B^{\pi_e,\tilde{P}_e}(s,a)$ for state-action pair $s,a$ in epoch $e$ is upper bounded as*

$$B^{\pi_e,\tilde{P}_e}(s,a) \le \min\left\{2, \sqrt{\frac{14S\log(2AT)}{1 \vee N_e(s,a)}}\right\}\|\tilde{h}(\cdot)\|_\infty \tag{92}$$

*Proof.* Starting with the definition of Bellman error in Equation (20), we get

$$B^{\pi_e,\tilde{P}_e}(s,a) = \lim_{\gamma \to 1} \left( Q_\gamma^{\pi_e,\tilde{P}_e}(s,a) - \left( r(s,a) + \gamma \sum_{s' \in \mathcal{S}} P(s'|s,a) V_\gamma^{\pi_e,\tilde{P}_e} \right) \right) \tag{93}$$

$$= \lim_{\gamma \to 1} \left( \left( r(s,a) + \gamma \sum_{s' \in \mathcal{S}} \tilde{P}_e(s'|s,a) V_\gamma^{\pi_e,\tilde{P}_e}(s') \right) \right.$$

$$\left. - \left( r(s,a) + \gamma \sum_{s' \in \mathcal{S}} P(s'|s,a) V_\gamma^{\pi_e,\tilde{P}_e}(s') \right) \right) \tag{94}$$

$$= \lim_{\gamma \to 1} \gamma \sum_{s' \in \mathcal{S}} \left( \tilde{P}_e(s'|s,a) - P(s'|s,a) \right) V_\gamma^{\pi_e,\tilde{P}_e}(s') \tag{95}$$

$$= \lim_{\gamma \to 1} \gamma \left( \sum_{s' \in \mathcal{S}} \left( \tilde{P}_e(s'|s,a) - P(s'|s,a) \right) V_\gamma^{\pi_e,\tilde{P}_e}(s') + V_\gamma^{\pi_e,\tilde{P}_e}(s) - V_\gamma^{\pi_e,\tilde{P}_e}(s) \right) \tag{96}$$

$$= \lim_{\gamma \to 1} \gamma \left( \sum_{s' \in \mathcal{S}} \left( \tilde{P}_e(s'|s,a) - P(s'|s,a) \right) V_\gamma^{\pi_e,\tilde{P}_e}(s') \right.$$

$$\left. - \sum_{s' \in \mathcal{S}} \tilde{P}_e(s'|s,a) V_\gamma^{\pi_e,\tilde{P}_e}(s) + \sum_{s' \in \mathcal{S}} P(s'|s,a) V_\gamma^{\pi_e,\tilde{P}_e}(s) \right) \tag{97}$$

$$= \lim_{\gamma \to 1} \gamma \left( \sum_{s' \in \mathcal{S}} \left( \tilde{P}_e(s'|s,a) - P(s'|s,a) \right) \left( V_\gamma^{\pi_e,\tilde{P}_e}(s') - V_\gamma^{\pi_e,\tilde{P}_e}(s) \right) \right) \tag{98}$$

$$= \left( \sum_{s' \in \mathcal{S}} \left( \tilde{P}_e(s'|s,a) - P(s'|s,a) \right) \lim_{\gamma \to 1} \gamma \left( V_\gamma^{\pi_e,\tilde{P}_e}(s') - V_\gamma^{\pi_e,\tilde{P}_e}(s) \right) \right) \tag{99}$$

$$= \left( \sum_{s' \in \mathcal{S}} \left( \tilde{P}_e(s'|s,a) - P(s'|s,a) \right) \tilde{h}(s') \right) \tag{100}$$

$$\leq \left\| \left( \tilde{P}_e(\cdot|s,a) - P(\cdot|s,a) \right) \right\|_1 \|\tilde{h}(\cdot)\|_\infty \tag{101}$$

$$\leq \sqrt{\frac{14 S \log(2At)}{1 \vee N_e(s,a)}} \|\tilde{h}(\cdot)\|_\infty \tag{102}$$

$$\leq \sqrt{\frac{14 S \log(2AT)}{1 \vee N_e(s,a)}} \|\tilde{h}(\cdot)\|_\infty \tag{103}$$

where Equation (95) comes from the assumption that the rewards are known to the agent. Equation (99) follows from the fact that the difference between value function at two states is bounded. Equation (100) comes from the definition of bias term (Puterman, 2014). Equation (101) follows from Hölder's inequality. In Equation (102), $\|\tilde{h}(\cdot)\|_\infty$ is the bias span of the MDP with transition probabilities $\tilde{P}_e$ for policy $\pi_e$. Also, the $\ell_1$ norm of probability vector is bounded using Lemma F.1 for start time $t_e$ of epoch $e$. $\qquad\square$

Additionally, note that the $\ell_1$ norm in Equation (101) is bounded by 2. Thus the Bellman error is loose upper bounded by $2\|\tilde{h}(\cdot)\|_\infty$ for all state-action pairs.

Note that we have converted the difference of average rewards into the average Bellman error. Also, we have bounded the Bellman error of a state-action pair. We now want to bound the average Bellman error of an epoch using the realizations of Bellman error at state-action pairs visited in an epoch. For this, we present the following lemma.

**Lemma D.4.** *With probability at least $1 - 1/T^6$, the cumulative expected Bellman error is bounded as:*

$$\sum_{e=1}^{E}(t_{e+1} - t_e)\mathbb{E}_{\pi_e,P}\left[B^{\pi_e,\tilde{P}_e}(s,a)\right] \leq \sum_{e=1}^{E}\sum_{t=t_e}^{t_{e+1}-1} B^{\pi_e,\tilde{P}_e}(s_t,a_t) + 4T_M\sqrt{7T\log(T)} \tag{104}$$

*Proof.* Let $\mathcal{F}_t = \{s_1, a_1, \cdots, s_t, a_t\}$ be the filtration generated by the running the algorithm for $t$ time-steps. Note that conditioned on filtration $\mathcal{F}_{t_e-1}$ the two expectations $\mathbb{E}_{s,a\sim\pi_e,P}[\cdot]$ and $\mathbb{E}_{s,a\sim\pi_e,P}[\cdot|\mathcal{F}_{t_e-1}]$ are not equal as the former is the expected value of the long-term state distribution and the latter is the long-term state distribution condition on initial state $s_{t_e-1}$. We now use Assumption 3.1 to obtain the following set of inequalities.

$$\mathbb{E}_{(s,a)\sim\pi_e,P}[B^{\pi_e,\tilde{P}_e}(s,a)] = \mathbb{E}_{(s,a)\sim\pi_e,P}[B^{\pi_e,\tilde{P}_e}(s,a)] \pm \mathbb{E}_{(s_t,a_t)\sim\pi_e,P}[B^{\pi_e,\tilde{P}_e}(s_t,a_t)|\mathcal{F}_{t_e-1}] \tag{105}$$

$$= \mathbb{E}_{(s_t,a_t)\sim\pi_e,P}[B^{\pi_e,\tilde{P}_e}(s_t,a_t)|\mathcal{F}_{t_e-1}]$$
$$+ \left(\mathbb{E}_{(s,a)\sim\pi_e,P}[B^{\pi_e,\tilde{P}_e}(s,a)] - \mathbb{E}_{(s_t,a_t)\sim\pi_e,P}[B^{\pi_e,\tilde{P}_e}(s_t,a_t)|H_{t_e-1}]\right) \tag{106}$$

$$\leq \mathbb{E}_{(s_t,a_t)\sim\pi_e,P}[B^{\pi_e,\tilde{P}_e}(s_t,a_t)|\mathcal{F}_{t_e-1}]$$
$$+ 2\|\tilde{h}(\cdot)\|_\infty \sum_{s,a}\left|\pi_e(a|s)d_{\pi_e}(s) - \pi_e(a|s)P_{\pi,s_{t_e-1}}^{t-t_e+1}(s)\right| \tag{107}$$

$$\leq \mathbb{E}_{(s_t,a_t)\sim\pi_e,P}[B^{\pi_e,\tilde{P}_e}(s_t,a_t)|\mathcal{F}_{t_e-1}]$$
$$+ 2\|\tilde{h}(\cdot)\|_\infty \sum_{s,a}\pi(a|s)\left|d_{\pi_e}(s) - P_{\pi,s_{t_e-1}}^{t-t_e+1}(s)\right| \tag{108}$$

$$\leq \mathbb{E}_{(s_t,a_t)\sim\pi_e,P}[B^{\pi_e,\tilde{P}_e}(s_t,a_t)|\mathcal{F}_{t_e-1}]$$
$$+ 2\|\tilde{h}(\cdot)\|_\infty \sum_{s,a}\pi(a|s)\|d_{\pi_e} - P_{\pi,s_{t_e-1}}^{t-t_e+1}\|_{TV} \tag{109}$$

$$\leq \mathbb{E}_{(s_t,a_t)\sim\pi_e,P}[B^{\pi_e,\tilde{P}_e}(s_t,a_t)|\mathcal{F}_{t_e-1}]$$
$$+ 2\|\tilde{h}(\cdot)\|_\infty \sum_{s,a}\pi(a|s)C\rho^{t-t_e} \tag{110}$$

$$= \mathbb{E}_{(s_t,a_t)\sim\pi_e,P}[B^{\pi_e,\tilde{P}_e}(s_t,a_t)|\mathcal{F}_{t_e-1}] + 2CS\|\tilde{h}(\cdot)\|_\infty\rho^{t-t_e} \tag{111}$$

where Equation 107 comes from Assumption 3.1 for running policy $\pi_e$ starting from state $s_{t_e-1}$ for $t - t_e + 1$ steps and from Lemma 5.3. Equation (111) follows from bounding the total-variation distance for all states and from the fact that $\sum_a \pi(a|s) = 1$.

Using this, and the fact that $\mathbb{E}_{\pi_e,P}\left[B^{\pi_e,\tilde{P}_e}(s_t,a_t)|\mathcal{F}_{t_e-1}\right] - B^{\pi_e,\tilde{P}_e}(s_t,a_t)$ forms a Martingale difference sequence conditioned on filtration $\mathcal{F}_{t-1}$ with $|\mathbb{E}_{\pi_e,P}\left[B^{\pi_e,\tilde{P}_e}(s_t,a_t)|\mathcal{F}_{t-1}\right] - B^{\pi_e,\tilde{P}_e}(s_t,a_t)| \leq 4\|\tilde{h}(\cdot)\|_\infty$, we can use Azuma-Hoeffding inequality to bound the summation as

$$\sum_{e=1}^{E}(t_{e+1} - t_e)\mathbb{E}_{\pi_e,P}\left[B^{\pi_e,\tilde{P}_e}(s,a)\right]$$

$$= \sum_{e=1}^{E}\left((t_{e+1} - t_e)\mathbb{E}_{\pi_e,P}\left[B^{\pi_e,\tilde{P}_e}(s_t,a_t)|\mathcal{F}_{t_e-1}\right] + \sum_{t=t_e}^{t_{e+1}-1} 2CS\|\tilde{h}(\cdot)\|_\infty\rho^{t-t_e}\right) \tag{112}$$

$$\leq \sum_{e=1}^{E} \left( \sum_{t=t_e}^{t_{e+1}-1} \mathbb{E}_{\pi_e, P}\left[ B^{\pi_e, \tilde{P}_e}(s_t, a_t) | \mathcal{F}_{t_e-1} \right] + \frac{2CS\|\tilde{h}(\cdot)\|_{\infty}}{1-\rho} \right) \tag{113}$$

$$\leq \sum_{e=1}^{E} \sum_{t=t_e}^{t_{e+1}-1} B^{\pi_e, \tilde{P}_e}(s_t, a_t) + 4\|\tilde{h}(\cdot)\|_{\infty} \sqrt{7T \log 2T} + \frac{2CES\|\tilde{h}(\cdot)\|_{\infty}}{1-\rho} \tag{114}$$

where Eq. (114) comes from the Azuma-Hoefdding's inequality with probability at least $1 - T^{-6}$. $\qquad\square$

## D.4 Bounding the term $\|\tilde{h}(\cdot)\|_{\infty}$

Note that we have $\lambda_{\pi_e}^{\tilde{P}_e} > \lambda_{\pi_e}^{P'}$ for all $P'$ in the confidence set.

**Lemma D.5.** *For a MDP with rewards $r(s,a)$ and transition probabilities $\tilde{P}_e$, using policy $\pi_e$, the difference of bias of any two states $s$, and $s'$ is bounded as $\tilde{h}(s) - \tilde{h}(s') \leq T_M \ \forall \ s, s' \in \mathcal{S}$.*

*Proof.* Note that $\lambda_{\pi_e}^{\tilde{P}_e} \geq \lambda_{\pi_e}^{P'}$ for all $P'$ in the confidence set. Now, consider the following Bellman equation

$$\tilde{h}(s) = r_{\pi_e}(s,a) - \lambda_{\pi_e}^{\tilde{P}_e} + (P_{\pi_e, e}(\cdot|s))^T \tilde{h} \ = T\tilde{h}(s)$$

where $r_{\pi_e}(s) = \sum_a \pi_e(a|s) r(s,a)$ and $P_{\pi_e, e}(s'|s) = \sum_a \pi(a|s) \tilde{P}_e(s'|s,a)$.

Consider two states $s, s' \in \mathcal{S}$. Also, let $\tau = \min\{t \geq 1 : s_t = s', s_1 = s\}$ be a random variable. With $P_{\pi_e}(\cdot|s) = \sum_a \pi_e(a|s) P(s'|s,a)$, we also define another operator,

$$\bar{T}h(s) = (\min_{s,a} r(s,a) - \lambda_{\pi_e}^{\tilde{P}_e} + (P_{\pi_e}(\cdot|s))^T h) \mathbf{1}(s \neq s') + \tilde{h}(s') \mathbf{1}(s = s').$$

Note that $\bar{T}\tilde{h}(s) \leq T\tilde{h}(s) = \tilde{h}(s)$ for all $s$ since $\tilde{P}_e$ maximizes the reward $r$ over all the transition probabilities in the confidence set of Eq. (21) including the true transition probability $P$. Further, for any two vectors $u, v \in \mathbb{R}^S$ with $u(s) \geq v(s) \forall s$, we have $\bar{T}u \geq \bar{T}v$. Hence, we have $\bar{T}^n \tilde{h}(s) \leq \tilde{h}(s)$ for all $s$. Hence, we have

$$\tilde{h}(s) \geq \bar{T}^n(s) = \mathbb{E}\left[ -(\lambda_{\pi_e}^{\tilde{P}_e} - \min_{s,a} r(s,a))(n \wedge \tau) + \tilde{h}(s_{n \wedge \tau}) \right]$$

Taking limit as $n \to \infty$, we have $\tilde{h}(s) \geq \tilde{h}(s') - T_M$, thus completing the proof. $\qquad\square$

We are now ready to bound $R_2(T)$ using Lemma D.1, Lemma D.3, and Lemma D.4. We have the following set of equations:

$$R_2(T) \ = \ \frac{L}{T} \Big| \sum_{e=1}^{E} \sum_{t=t_e}^{t_{e+1}-1} \left( \lambda_{\pi_e}^{\tilde{P}_e} - \lambda_{\pi_e}^{P} \right) \Big| \tag{115}$$

$$= \ \frac{L}{T} \Big| \sum_{e=1}^{E} \sum_{t=t_e}^{t_{e+1}-1} \sum_{s,a} \rho_{\pi_e}^{P} B^{\pi_e, \tilde{P}_e}(s,a) \Big| \tag{116}$$

$$\leq \ \frac{L}{T} \Big| \sum_{e=1}^{E} \sum_{t=t_e}^{t_{e+1}-1} B^{\pi_e, \tilde{P}_e}(s_t, a_t) + 4T_M \sqrt{7T \log(2T)} + \frac{2CT_M SE}{1-\rho} \Big| \tag{117}$$

$$\leq \frac{L}{T}\Big|\sum_{e=1}^{E}\sum_{t=t_e}^{t_{e+1}-1} T_M\sqrt{\frac{14S\log(2AT)}{1\vee N_e(s,a)}} + 4T_M\sqrt{7T\log(2T)} + \frac{2CT_MSE}{1-\rho}\Big| \tag{118}$$

$$\leq \frac{L}{T}\Big|\sum_{e=1}^{E}\sum_{s,a}\nu_e(s,a)T_M\sqrt{\frac{14S\log(2AT)}{1\vee N_e(s,a)}} + 4T_M\sqrt{7T\log(2T)} + \frac{2CT_MSE}{1-\rho}\Big| \tag{119}$$

$$\leq \frac{L}{T}\Big|\sum_{s,a} T_M\sqrt{14SA\log(2AT)}\sum_{e=1}^{E}\frac{\nu_e(s,a)}{\sqrt{1\vee N_e(s,a)}} + 4T_M\sqrt{7T\log(2T)} + \frac{2CT_MSE}{1-\rho}\Big| \tag{120}$$

$$\leq \frac{L}{T}\Big|\sum_{s,a} T_M(\sqrt{2}+1)\sqrt{14SA\log(2AT)}\sqrt{N(s,a)} + 4T_M\sqrt{7T\log(2T)} + \frac{2CT_MSE}{1-\rho}\Big| \tag{121}$$

$$\leq \frac{L}{T}\Big|T_M(\sqrt{2}+1)\sqrt{14SA\log(2AT)}\sqrt{\Big(\sum_{s,a}1\Big)\Big(\sum_{s,a}N(s,a)\Big)}$$
$$+ 4T_M\sqrt{7T\log(2T)} + \frac{2CT_MSE}{1-\rho}\Big| \tag{122}$$

$$\leq \frac{L}{T}\Big|T_M(\sqrt{2}+1)\sqrt{14SA\log(2AT)}\sqrt{SAT} + 4T_M\sqrt{7T\log(2T)} + \frac{2CT_MSE}{1-\rho}\Big| \tag{123}$$

where Equation (116) follows from Lemma D.1, Equation (117) follows from Lemma D.4, and Equation (118) follows from Lemma D.4. Equation (121) follows from (Jaksch et al., 2010) and Equation (122) follows from Cauchy-Schwarz inequality.

## D.5 Bounding $R_3(T)$

Bounding $R_3(T)$ follows mostly similar to Lemma D.4. At each epoch, the agent visits states according to the occupancy measure $\rho^P_{\pi_e}$ and obtains the rewards. We bound the deviation of the observed visitations to the expected visitations to each state action pair in each epoch.

**Lemma D.6.** *With probability at least $1-1/T^6$, the difference between the observed rewards and the expected rewards is bounded as:*

$$\Big|\sum_{e=1}^{E}\sum_{t=t_e}^{t_{e+1}-1}\mathbb{E}_{\pi_e,P}\left[r(s,a)\right] - \sum_{e=1}^{E}\sum_{t=t_e}^{t_{e+1}-1}r(s_t,a_t)\Big| \leq 2\sqrt{7T\log(2T)} \tag{124}$$

*Proof.* We note that $\mathbb{E}_{\pi_e,P}\left[r(s,a)|\mathcal{F}_{t-1}\right] - r(s_t,a_t)$ is a Martingale difference sequence bounded by 2 because the rewards are bounded by 1. Hence, following the proof of Lemma D.4 we get the required result. $\square$

## D.6 Bounding the number of episodes $E$

The number of episodes $E$ of the UC-CURL algorithm are bounded by $1 + 2SA + SA\log(T/SA)$ from Proposition 18 of (Jaksch et al., 2010). We now bound the number of episodes for the modification of the algorithm as described in Section 5.2. We considered to trigger a new episode whenever $\nu_e(s,a)$ becomes $\max\{1,\bar{\nu}_{e-1}(s,a)+1\}$ where $\bar{\nu}_{e-1}$ is the number of visitations to $s,a$ which triggered a new epoch. In the following lemma, we show that the number of episodes are bounded by $O(1+\sqrt{2SAT})$ with this epoch trigger schedule.

**Lemma D.7.** *If the UC-CURL algorithm triggers a new epoch whenever $\nu_e(s,a) \geq \max\{1,\bar{\nu}_{e-1}(s,a)+1\}$ for any state-action pair $s,a$, the total number of epochs are bounded by $O(1+\sqrt{2SAT})$, where $\bar{\nu}_e(s,a) = \nu_e(s,a)\mathbf{1}\{\nu_e(s,a) = \bar{\nu}_{e-1}(s,a)+1\} + \bar{\nu}_{e-1}(s,a)(s,a)\mathbf{1}\{\nu_e(s,a) \neq \bar{\nu}_{e-1}(s,a)\}$ and $\nu_0(s,a) = \bar{\nu}_e(s,a) = 1$ for all $s,a$.*

*Proof.* Let $N(s, a)$ be the number visitations to state-action pair $s, a$ and $K(s, a)$ be the total number of epochs triggered when the trigger condition is met for state action pair $s, a$. Hence, we have

$$N(s, a) = \sum_{e=1}^{E} \nu_e(s, a) \tag{125}$$

$$\geq \sum_{e: \nu_e(s,a) = \bar{\nu}_{e-1}+1} \nu_e(s, a) \tag{126}$$

$$\geq \frac{K(s, a)(K(s, a) + 1)}{2} \geq \frac{K^2(s, a)}{2}, \tag{127}$$

where considering only epoch triggers for $s, a$ gives Equation (126). Equation (127) is obtained from the fact that $\bar{\nu}_e(s, a) = \nu_e(s, a) = \bar{\nu}_{e-1}(s, a) + 1$ which gives $N_{e+1}(s, a) = N_e(s, a) + \bar{\nu}_{e+1}(s, a) = N_e(s, a) + \bar{\nu}_e(s, a) + 1 = e(e + 1)/2$.

Now, we have the following,

$$T = \sum_{s,a} N(s, a) \tag{128}$$

$$\geq \sum_{s,a} \frac{K^2(s, a)}{2} \tag{129}$$

$$= \frac{SA}{2SA} \sum_{s,a} K^2(s, a) \tag{130}$$

$$\geq \frac{SA}{2} \left( \frac{1}{SA} \sum_{s,a} K(s, a) \right)^2 \tag{131}$$

where Equation (131) is obtained from the convexity of $x^2$. Hence, we have,

$$\sum_{s,a} K(s, a) \leq SA \sqrt{\frac{2T}{SA}} = \sqrt{2SAT} \tag{132}$$

Further, the first epoch is triggered when the algorithm starts. Hence we have $E = 1 + \sum_{s,a} K(s, a) = 1 + \sqrt{2SAT}$. $\qquad \square$

## E    Bounding Constraint Violations

To bound the constraint violations $C(T)$, we break it into multiple components. We can then bound these components individually.

### E.1 Constraint breakdown

We first break down our constraint violations into multiple parts which will help us bound the constraint violations.

$$C(T) = \left( g \left( \frac{1}{T} \sum_{t=1}^{T} c_1(s_t, a_t), \cdots, \frac{1}{T} \sum_{t=1}^{T} c_d(s_t, a_t) \right) \right)_+ \tag{133}$$

$$= \left( g \left( \frac{1}{T} \sum_{t=1}^{T} c_1(s_t, a_t), \cdots, \frac{1}{T} \sum_{t=1}^{T} c_d(s_t, a_t) \right) + \frac{1}{T} \sum_{e=1}^{E} T_e g \left( \zeta_{\pi_e}^{\tilde{P}_e}(1), \cdots, \zeta_{\pi_e}^{\tilde{P}_e}(d) \right) \right.$$

$$\left. - \frac{1}{T} \sum_{e=1}^{E} T_e g \left( \zeta_{\pi_e}^{\tilde{P}_e}(1), \cdots, \zeta_{\pi_e}^{\tilde{P}_e}(d) \right) \right)_+ \tag{134}$$

$$\leq \left( g \left( \frac{1}{T} \sum_{t=1}^{T} c_1(s_t, a_t), \cdots, \frac{1}{T} \sum_{t=1}^{T} c_d(s_t, a_t) \right) - \frac{1}{T} \sum_{e=1}^{E} T_e g \left( \zeta_{\pi_e}^{\tilde{P}_e}(1), \cdots, \zeta_{\pi_e}^{\tilde{P}_e}(d) \right) - \frac{1}{T} \sum_{e=1}^{E} T_e \epsilon_e \right)_+ \tag{135}$$

$$\leq \left( g \left( \frac{1}{T} \sum_{t=1}^{T} c_1(s_t, a_t), \cdots, \frac{1}{T} \sum_{t=1}^{T} c_d(s_t, a_t) \right) - g \left( \frac{1}{T} \sum_{e=1}^{E} T_e \zeta_{\pi_e}^{\tilde{P}_e}(1), \cdots, \frac{1}{T} \sum_{e=1}^{E} T_e \zeta_{\pi_e}^{\tilde{P}_e}(d) \right) - \frac{1}{T} \sum_{e=1}^{E} T_e \epsilon_e \right)_+ \tag{136}$$

$$\leq \left( L \sum_{i=1}^{d} \Big| \frac{1}{T} \sum_{e=1}^{E} \sum_{t=t_e}^{t_{e+1}-1} \left( c_i(s_t, a_t) - \zeta_{\pi_e}^{\tilde{P}_e}(i) \right) \Big| - \frac{1}{T} \sum_{e=1}^{E} T_e \epsilon_e \right)_+ \tag{137}$$

$$\leq \left( L \sum_{i=1}^{d} \Big| \frac{1}{T} \sum_{e=1}^{E} \sum_{t=t_e}^{t_{e+1}-1} \left( c_i(s_t, a_t) - \zeta_{\pi_e}^{P}(i) + \zeta_{\pi_e}^{P}(i) - \zeta_{\pi_e}^{\tilde{P}_e}(i) \right) \Big| - \frac{1}{T} \sum_{e=1}^{E} T_e \epsilon_e \right)_+ \tag{138}$$

$$\leq \left( \frac{L}{T} \sum_{i=1}^{d} \Big| \sum_{e=1}^{E} \sum_{t=t_e}^{t_{e+1}-1} \left( c_i(s_t, a_t) - \zeta_{\pi_e}^{P}(i) \right) \Big| + \frac{L}{T} \sum_{i=1}^{d} \Big| \sum_{e=1}^{E} \sum_{t=t_e}^{t_{e+1}-1} \left( \zeta_{\pi_e}^{P}(i) - \zeta_{\pi_e}^{\tilde{P}_e}(i) \right) \Big| - \frac{1}{T} \sum_{e=1}^{E} T_e \epsilon_e \right)_+ \tag{139}$$

$$\leq (C_3(T) + C_2(T) - C_1(T))_+ \tag{140}$$

where Equation (185) comes from the fact the policy $\pi_e$ is solution of a conservative optimization equation. Equation (186) comes from the convexity of the constraint $g(\cdot)$. Equation (187) follows from the Lipschitz assumption. The three terms in Equation (67) are now defined as:

$$C_1(T) = \frac{1}{T} \sum_{e=1}^{E} T_e \epsilon_e \tag{141}$$

$C_1(T)$ denotes the gap left by playing the policy for $\epsilon_e$-tight optimization problem on the optimistic MDP.

$$C_2(T) = \frac{L}{T} \sum_{i=1}^{d} \Big| \sum_{e=1}^{E} \sum_{t=t_e}^{t_{e+1}-1} \left( \zeta_{\pi_e}^{P}(i) - \zeta_{\pi_e}^{\tilde{P}_e}(i) \right) \Big| \tag{142}$$

$C_2(T)$ denotes the difference between long-term average costs incurred by playing the policy $\pi_e$ on the true MDP with transitions $P$ and the optimistic MDP with transitions $\tilde{P}$. This term is bounded similar to the bound of $R_2(T)$.

$$C_3(T) = \frac{L}{T} \sum_{i=1}^{d} \Big| \sum_{e=1}^{E} \sum_{t=t_e}^{t_{e+1}-1} \left( c_i(s_t, a_t) - \zeta_{\pi_e}^{P}(i) \right) \Big| \tag{143}$$

$C_3(T)$ denotes the difference between long-term average costs incurred by playing the policy $\pi_e$ on the true MDP with transitions $P$ and the realized costs. This term is bounded similar to the bound of $R_3(T)$.

### E.2   Bounding $C_1(T)$

Note that $C_1(T)$ allows us to violate constraints by not having the knowledge of the true MDP and allowing deviations of incurred costs from the expected costs. We now want to lower bound $C_1$ to allow us sufficient slackness. With this idea, we have the following set of equations.

$$C_1(T) = \frac{1}{T} \sum_{e=1}^{E} \sum_{t=t_e}^{t_{e+1}-1} \epsilon_e \tag{144}$$

$$= \frac{1}{T} \sum_{e=1}^{E} \sum_{t=t_e}^{t_{e+1}-1} K \sqrt{\frac{\log t_e}{t_e}} \tag{145}$$

$$\geq K \frac{1}{T} \sum_{e=E'}^{E} \sum_{t=t_e}^{t_{e+1}-1} \sqrt{\frac{\log(T/4)}{t_e}} \tag{146}$$

$$\geq K \frac{1}{T} \sum_{e=E'}^{E} \sum_{t=t_e}^{t_{e+1}-1} \sqrt{\frac{\log(T/4)}{T}} \tag{147}$$

$$= K \frac{1}{T} \left( T - t_{E'} \right) \sqrt{\frac{\log(T/4)}{T}} \tag{148}$$

$$\geq K \frac{1}{2} \sqrt{\frac{\log(T/4)}{T}} \tag{149}$$

$$\geq K \frac{1}{4} \sqrt{\frac{\log T}{T}} \tag{150}$$

where $E'$ is some epoch for which $T/4 \leq t_{E'} < T/2$.

### E.3   Bounding $C_2(T)$, and $C_3(T)$

We note that costs incurred in $C_2(T)$, and $C_3(T)$ follows the same bound as $R_2(T)$ and $R_3(T)$ respectively. Thus, replacing $r$ with $c$, we obtain constraint violations because of imperfect system knowledge and system stochastics as $\tilde{O}(LdT_M S\sqrt{A/T})$.

Summing the three terms gives the required bound and choosing $K = \Theta(LdT_M S\sqrt{A})$ gives the required bound on constraint violations.

## F   Concentration bound results

We want to bound the deviation of the estimates of the estimated transition probabilities of the Markov Decision Processes $\mathcal{M}$. For that we use $\ell_1$ deviation bounds from (Weissman et al., 2003). Consider, the following event,

$$\mathcal{E}_t = \left\{ \|\hat{P}(\cdot|s,a) - P(\cdot|s,a)\|_1 \leq \sqrt{\frac{14S \log(2AT)}{\max\{1, n(s,a)\}}} \forall (s,a) \in \mathcal{S} \times \mathcal{A} \right\} \tag{151}$$

where $n = \sum_{t'=1}^{t} \mathbf{1}_{\{s_{t'}=s, a_{t'}=a\}}$. Then, we have the following result:

**Lemma F.1.** *The probability that the event $\mathcal{E}_t$ fails to occur us upper bounded by $\frac{1}{20t^6}$.*

*Proof.* From the result of (Weissman et al., 2003), the $\ell_1$ distance of a probability distribution over $S$ events with $n$ samples is bounded as:

$$\mathbb{P}\left(\|P(\cdot|s,a) - \hat{P}(\cdot|s,a)\|_1 \geq \epsilon\right) \leq (2^S - 2)\exp\left(-\frac{n\epsilon^2}{2}\right) \leq (2^S)\exp\left(-\frac{n\epsilon^2}{2}\right) \tag{152}$$

This, for $\epsilon = \sqrt{\frac{2}{n(s,a)}\log(2^S 20SAt^7)} \leq \sqrt{\frac{14S}{n(s,a)}\log(2At)} \leq \sqrt{\frac{14S}{n(s,a)}\log(2AT)}$ gives,

$$\mathbb{P}\left(\|P(\cdot|s,a) - \hat{P}(\cdot|s,a)\|_1 \geq \sqrt{\frac{14S}{n(s,a)}\log(2At)}\right) \leq (2^S)\exp\left(-\frac{n(s,a)}{2}\frac{2}{n(s,a)}\log(2^S 20SAt^7)\right) \tag{153}$$

$$= 2^S \frac{1}{2^S 20SAt^7} \tag{154}$$

$$= \frac{1}{20ASt^7} \tag{155}$$

We sum over the all the possible values of $n(s,a)$ till $t$ time-step to bound the probability that the event $\mathcal{E}_t$ does not occur as:

$$\sum_{n(s,a)=1}^{t} \frac{1}{20SAt^7} \leq \frac{1}{20SAt^6} \tag{156}$$

Finally, summing over all the $s, a$, we get,

$$\mathbb{P}\left(\|P(\cdot|s,a) - \hat{P}(\cdot|s,a)\|_1 \geq \sqrt{\frac{14S}{n(s,a)}\log(2At)} \; \forall s, a\right) \leq \frac{1}{20t^6} \tag{157}$$

$\square$

The second lemma is the Azuma-Hoeffding's inequality, which we use to bound Martingale difference sequences.

**Lemma F.2** (Azuma-Hoeffding's Inequality). *Let $X_1, \cdots, X_n$ be a Martingale difference sequence such that $|X_i| \leq c$ for all $i \in \{1, 2, \cdots, n\}$, then,*

$$\mathbb{P}\left(|\sum_{i=1}^{n} X_i| \geq \epsilon\right) \leq 2\exp\left(-\frac{\epsilon^2}{2nc^2}\right) \tag{158}$$

# G   Posterior Sampling Algorithm

Note that in the UC-CURL algorithm, the agent solves for an optimistic policy. This convex optimization problem may be computationally intensive with $O(S^2 A)$ additional variables and $O(SA)$ additional constraints. We now present the posterior sampling version of the UC-CURL algorithm which reduces this computational complexity by sampling the transition probabilities from the updated posterior. The posterior sampling algorithm is based on Lemma 1 of (Osband et al., 2013), which we state formally here.

**Lemma G.1.** *[Posterior Sampling] If $h$ is the distribution of $\mathcal{M}$ then, for any $\sigma(F_{t_e})$-measurable function $g$,*

$$\mathbb{E}\left[g(\mathcal{M})|F_{t_e}\right] = \mathbb{E}\left[g(\mathcal{M}_e)|F_{t_e}\right] \tag{159}$$

*where $\mathcal{M}_e$ is the MDP sampled at the beginning of the epoch $e$ at time-step $t_e$.*

---

**Algorithm 2** PS-CURL

---

**Parameters**: $K$

**Input**: $S$, $A$, $r$, $d$, $c_i$ $\forall$ $i \in [d]$

1: Let $t = 1$, $e = 1$, $\epsilon_e = K\sqrt{\frac{\ln t}{t}}$
2: $\nu_e(s,a) = 0$, $N_e(s,a) = 0$ $\forall$ $s, a$
3: Solve for policy $\pi_e$ using Eq. (164)
4: **for** $t \in \{1, 2, \cdots\}$ **do**
5:     Observe $s_t$, and play $a_t \sim \pi_e(\cdot|s_t)$
6:     Observe $s_{t+1}$, $r(s_t, a_t)$ and $c_i(s_t, a_t)$ $\forall$ $i \in [d]$
7:     $\nu_e(s_t, a_t) = \nu_e(s_t, a_t) + 1$
8:     **if** $\nu_e(s,a) = \max\{1, N_e(s,a)\}$ for any $s, a$ **then**
9:         **for** $(s,a) \in \mathcal{S} \times \mathcal{A}$ **do**
10:             $N_{e+1}(s,a) = N_e(s,a) + \nu_e(s,a)$
11:             $e = e + 1$, $\nu_e(s,a) = 0$
12:         **end for**
13:         $\epsilon_e = K\sqrt{\frac{\ln t}{t}}$
14:         $\tilde{P}_e \sim h(\cdot|\mathcal{H}_t)$
15:         Solve for policy $\pi_e$ using Eq. (164)
16:     **end if**
17: **end for**

---

We now present our posterior sampling based PS-CURL algorithm described in Algorithm 2. Similar to the UC-CURL algorithm, the PS-CURL algorithm proceeds in epochs. At each epoch $e$, the agent samples $\tilde{P}_e \sim h(\cdot|\mathcal{F}_{t_e})$, it can solve the following optimization problem for the optimal feasible policy.

$$\max_{\rho(s,a)} f\big(\sum\nolimits_{s,a} r(s,a)\rho_e(s,a)\big) \tag{160}$$

with the following set of constraints,

$$\sum\nolimits_{s,a} \rho_e(s,a) = 1, \quad \rho_e(s,a) \geq 0 \tag{161}$$

$$\sum\nolimits_{a \in \mathcal{A}} \rho_e(s',a) = \sum\nolimits_{s,a} \tilde{P}_e(s'|s,a)\rho_e(s,a) \tag{162}$$

$$g\big(\sum\nolimits_{s,a} c_1(s,a)\rho_e(s,a), \cdots, \sum\nolimits_{s,a} c_d(s,a)\rho_e(s,a)\big) \leq -\epsilon_e \tag{163}$$

for all $s' \in \mathcal{S}$, $\forall$ $s \in \mathcal{S}$, and $\forall$ $a \in \mathcal{A}$. Using the solution for $\rho_e$ for $\epsilon_e$-tight optimization equation for the optimistic MDP, we obtain the optimal conservative policy for epoch $e$ as:

$$\pi_e(a|s) = \frac{\rho_e(s,a)}{\sum_{b \in \mathcal{A}} \rho_e(s,b)} \forall \ s, a \tag{164}$$

For the UC-CURL algorithm, the true MDP lies in the confidence interval with high probability, and hence the solution of the optimization problem was guaranteed. However, the same is not true for the MDP with sampled transition probabilities. We want the existence of a policy $\pi_e$ such that Equation (163) holds. We obtain the condition for existence of such a policy in the following lemma. To obtain the lemma, we first state a tighter Slater assumption as:

**Assumption G.2.** There exists a policy $\pi$, and constants $delta > LdST_M\sqrt{(A\log T)/T} + (CSA\log T)/(T(1-\rho))$ and $\Gamma > Ld\left(2ST_M\sqrt{14A\log AT/T^{1/3}} + CST_M/((1-\rho)T^{1/3})\right)$ such that

$$g\left(\zeta_\pi^{P,1}, \cdots, \zeta_\pi^{P,K_2}\right) \leq -\delta - \Gamma \tag{165}$$

**Lemma G.3.** *If there exists a policy $\pi$, such that*

$$g\left(\zeta_\pi^P(1), \cdots, \zeta_\pi^P(d)\right) \leq -\delta - \Gamma, \tag{166}$$

*and there exists episodes $e$ and $e+1$ with start timesteps $t_e$ and $t_{e+1}$ respectively satisfying $t_{e+1} - t_e \geq T^{1/3}$, then for $\|\tilde{P}_e(\cdot|s,a) - P(\cdot|s,a)\|_1 \leq \sqrt{\frac{14S\log(2At)}{N_e(s,a)}}$ the policy $\pi$ satisfies,*

$$g\left(\zeta_\pi^{\tilde{P}_e}(1), \cdots, \zeta_\pi^{\tilde{P}_e}(d)\right) \leq -\delta. \tag{167}$$

*Proof.* We start with the Lipschitz assumption (Assumption 3.4) to obtain,

$$|g\left(\zeta_\pi^{\tilde{P}_e}(1), \cdots, \zeta_\pi^{\tilde{P}_e}(d)\right) - g\left(\zeta_\pi^P(1), \cdots, \zeta_\pi^P(d)\right)| \leq Ld\max_i |\zeta_\pi^{\tilde{P}_e}(i) - \zeta_\pi^P(i)| \tag{168}$$

$$\implies g\left(\zeta_\pi^{\tilde{P}_e}(1), \cdots, \zeta_\pi^{\tilde{P}_e}(d)\right) \leq Ld\max_i |\zeta_\pi^{\tilde{P}_e}(i) - \zeta_\pi^P(i)| + g\left(\zeta_\pi^P(1), \cdots, \zeta_\pi^P(d)\right) \tag{169}$$

where Equation (169) is obtained by choosing the sign of modulo in the previous equation. We now bound the term $|\zeta_\pi^{\tilde{P}_e} - \zeta_\pi^P|$ using Bellman error. We have,

$$\zeta_\pi^{\tilde{P}_e}(i) - \zeta_\pi^P(i) = \sum_{s,a} \rho_\pi^P B_i^{\pi_e, \tilde{P}_e}(s,a) = \mathbb{E}\left[B^{\pi_e, \tilde{P}_e}(s,a)\right] \tag{170}$$

where $B_i^{\pi_e, \tilde{P}_e}(s,a)$ is the Bellman error for cost $i$. We bound the expectation using Azuma-Hoeffding's inequality as follows:

$$\mathbb{E}\left[B^{\pi_e, \tilde{P}_e}(s,a)\right] = \mathbb{E}\left[B^{\pi_e, \tilde{P}_e}(s_t, a_t)|\mathcal{F}_{t_e-1}\right] + C\rho^{t-t_e} \tag{171}$$

$$= \frac{1}{t_{e+1} - t_e} \sum_{t_e}^{t_{e+1}-1}\left(\mathbb{E}\left[B^{\pi_e, \tilde{P}_e}(s_t, a_t)|\mathcal{F}_{t_e-1}\right] + C\rho^{t-t_e}\right) \tag{172}$$

$$\leq \frac{1}{t_{e+1} - t_e} \sum_{t_e}^{t_{e+1}-1}\left(\mathbb{E}\left[B^{\pi_e, \tilde{P}_e}(s_t, a_t)|\mathcal{F}_{t_e-1}\right]\right) + \frac{CS\|\tilde{h}(\cdot)\|_\infty}{(1-\rho)(t_{e+1} - t_e)} \tag{173}$$

$$\leq \frac{1}{t_{e+1} - t_e}\left(\|\tilde{h}\|_\infty\sqrt{14S\log AT}\sum_{s,a}\frac{\nu_e(s,a)}{\sqrt{N_e(s,a)}} + 4\|\tilde{h}(\cdot)\|_\infty\sqrt{7(t_{e+1}-t_e)\log(t_{e+1}-t_e)}\right)$$
$$+ \frac{CS\|\tilde{h}(\cdot)\|_\infty}{(1-\rho)(t_{e+1}-t_e)} \tag{174}$$

$$\leq \frac{1}{t_{e+1} - t_e}\left(\|\tilde{h}\|_\infty\sqrt{14S\log AT}\sum_{s,a}\sqrt{\nu_e(s,a)} + 4\|\tilde{h}(\cdot)\|_\infty\sqrt{7(t_{e+1}-t_e)\log(t_{e+1}-t_e)}\right)$$
$$+ \frac{CS\|\tilde{h}(\cdot)\|_\infty}{(1-\rho)(t_{e+1}-t_e)} \tag{175}$$

$$\leq \frac{1}{t_{e+1} - t_e}\left(\|\tilde{h}\|_\infty S\sqrt{14A\log AT}\sqrt{\sum_{s,a}\nu_e(s,a)} + 4\|\tilde{h}(\cdot)\|_\infty\sqrt{7(t_{e+1}-t_e)\log(t_{e+1}-t_e)}\right)$$
$$+ \frac{CS\|\tilde{h}(\cdot)\|_\infty}{(1-\rho)(t_{e+1}-t_e)} \tag{176}$$

$$\leq \frac{1}{t_{e+1} - t_e} \left( \|\tilde{h}\|_\infty S \sqrt{14A \log AT} \sqrt{(t_{e+1} - t_e)} + 4\|\tilde{h}(\cdot)\|_\infty \sqrt{7(t_{e+1} - t_e) \log(t_{e+1} - t_e)} \right)$$

$$+ \frac{CS\|\tilde{h}(\cdot)\|_\infty}{(1 - \rho)(t_{e+1} - t_e)} \tag{177}$$

$$\leq \left( \|\tilde{h}\|_\infty S \sqrt{\frac{14A \log AT}{(t_{e+1} - t_e)}} + 4\|\tilde{h}(\cdot)\|_\infty \sqrt{\frac{7 \log(t_{e+1} - t_e)}{(t_{e+1} - t_e)}} \right) + \frac{CS\|\tilde{h}(\cdot)\|_\infty}{(1 - \rho)(t_{e+1} - t_e)} \tag{178}$$

where Equation (172) is obtained by summing both sides from $t = t_e$ to $t = t_{e+1}$. Equation (173) is obtained by summing over the geometric series with ratio $\rho$. Equation (174) comes from Lemma D.4. Equation (175) comes from the fact that $N_e(s, a) \geq \nu_e(s, a)$ for all $s, a$, and then replacing the lower bound of $N_e(s, a)$. Equation (176) follows from the Cauchy Schwarz inequality. Equation (177) follows from the fact that the epoch length $t_{e+1} - t_e$ is same as the number of visitations to all state action pairs in an epoch.

Combining Equation (178) with Equation (169), and bounding the $\|\tilde{h}(\cdot)\|_\infty$ term with $T_M$, we obtain the required result as follows:

$$g\left( \zeta_\pi^{\tilde{P}_e}(1), \cdots, \zeta_\pi^{\tilde{P}_e}(d) \right) \leq Ld \max_i |\zeta_\pi^{\tilde{P}_e}(i) - \zeta_\pi^P(i)| + g\left( \zeta_\pi^P(1), \cdots, \zeta_\pi^P(d) \right) \tag{179}$$

$$\leq Ld\Big( \left( T_M S \sqrt{\frac{14A \log AT}{(t_{e+1} - t_e)}} + 4T_M \sqrt{\frac{7 \log(t_{e+1} - t_e)}{(t_{e+1} - t_e)}} \right)$$

$$+ \frac{CT_M S}{(1 - \rho)(t_{e+1} - t_e)} \Big) + g\left( \zeta_\pi^P(1), \cdots, \zeta_\pi^P(d) \right) \tag{180}$$

$$\leq Ld\Big( \left( T_M S \sqrt{\frac{14A \log AT}{(t_{e+1} - t_e)}} + 4T_M \sqrt{\frac{7 \log(t_{e+1} - t_e)}{(t_{e+1} - t_e)}} \right)$$

$$+ \frac{CT_M S}{(1 - \rho)(t_{e+1} - t_e)} \Big) - \delta - \Gamma \tag{181}$$

$$\leq -\delta, \tag{182}$$

where Equation (182) is follows from the definition of $\Gamma$ in Assumption G.2 and $t_{e+1} - t_e \geq T^{1/3}$. $\qquad \square$

From Lemma G.3 we observe that for a tighter Slater condition on the true MDP, we can only guarantee a weaker Slater guarantee. However, we make that assumption to obtain the feasibility of the optimization problem in Equation (160).

The Bayesian regret of the PS-CURL algorithm is defined as follows:

$$\mathbb{E}[R(T)] = \mathbb{E}\left[ f\left(\lambda_{\pi^*}^P\right) - f\left(\sum_{t=1}^T r(s_t, a_t)/T\right) \right]$$

Similarly, we define Bayesian constraint violations, $C(T)$, as the expected gap between the constraint function and incurred and constraint bounds, or

$$\mathbb{E}[C(T)] = \mathbb{E}\left[ \left( g\left( \sum_{t=1}^T c_1(s_t, a_t)/T, \cdots, \sum_{t=1}^T c_1(s_t, a_t)/T \right) \right)_+ \right]$$

where $(x)_+ = \max(0, x)$.

Now, we can use Lemma G.1 to obtain $\mathbb{E}[f(\lambda_{\pi^*}^P)|\mathcal{F}_{t_e}] = \mathbb{E}[f(\lambda_{\pi_e}^{\tilde{P}_e})|\mathcal{F}_{t_e}]$ and $\mathbb{E}[f(\zeta_{\pi^*}^P)(i)|\mathcal{F}_{t_e}] = \mathbb{E}[f(\zeta_{\pi_e}^{\tilde{P}_e})(i)|\mathcal{F}_{t_e}] \; \forall \; i$, and follow the analysis similar to the analysis of Theorem 5.6 to obtain the required regret bounds.

### G.1 Bound on constraints

We now bound the constraint violations and prove that using a conservative policy. We can reduce the constraint violations to 0. We have:

$$C(T) = \left( g\left( \frac{1}{T}\sum_{t=1}^{T} c_1(s_t, a_t), \cdots, \frac{1}{T}\sum_{t=1}^{T} c_d(s_t, a_t) \right) \right)_{+} \tag{183}$$

$$= \left( g\left( \frac{1}{T}\sum_{t=1}^{T} c_1(s_t, a_t), \cdots, \frac{1}{T}\sum_{t=1}^{T} c_d(s_t, a_t) \right) - \frac{1}{T}\sum_{e=1}^{E} T_e g\left( \zeta_{\pi_e}^{\tilde{P}_e,1}, \cdots, \zeta_{\pi_e}^{\tilde{P}_e,K_2} \right) \right.$$

$$\left. + \frac{1}{T}\sum_{e=1}^{E} T_e g\left( \zeta_{\pi_e}^{\tilde{P}_e,1}, \cdots, \zeta_{\pi_e}^{\tilde{P}_e,K_2} \right) \right)_{+} \tag{184}$$

$$\leq \left( g\left( \frac{1}{T}\sum_{t=1}^{T} c_1(s_t, a_t), \cdots, \frac{1}{T}\sum_{t=1}^{T} c_d(s_t, a_t) \right) - \frac{1}{T}\sum_{e=1}^{E} T_e g\left( \zeta_{\pi_e}^{\tilde{P}_e,1}, \cdots, \zeta_{\pi_e}^{\tilde{P}_e,K_2} \right) + C_1 \right)_{+} \tag{185}$$

$$\leq \left( g\left( \frac{1}{T}\sum_{t=1}^{T} c_1(s_t, a_t), \cdots, \frac{1}{T}\sum_{t=1}^{T} c_d(s_t, a_t) \right) - g\left( \frac{1}{T}\sum_{e=1}^{E} T_e \zeta_{\pi_e}^{\tilde{P}_e,1}, \cdots, \frac{1}{T}\sum_{e=1}^{E} T_e \zeta_{\pi_e}^{\tilde{P}_e,K_2} \right) + C_1 \right)_{+} \tag{186}$$

$$\leq \left( L\sum_{k=1}^{K_2} \left| \frac{1}{T}\sum_{e=1}^{E}\sum_{t=t_e}^{t_{e+1}-1} \left( c^k(s_t, a_t) - \zeta_{\pi_e}^{\tilde{P}_e,k} \right) \right| + C_1 \right)_{+} \tag{187}$$

$$\leq \left( L\sum_{k=1}^{K_2} \left| \frac{1}{T}\sum_{e=1}^{E}\sum_{t=t_e}^{t_{e+1}-1} \left( c^k(s_t, a_t) - \zeta_{\pi_e}^{P,k} + \zeta_{\pi_e}^{P,k} - \zeta_{\pi_e}^{\tilde{P}_e,k} \right) \right| + C_1 \right)_{+} \tag{188}$$

$$\leq \left( \frac{L}{T}\sum_{k=1}^{K_2} \left| \sum_{e=1}^{E}\sum_{t=t_e}^{t_{e+1}-1} \left( c^k(s_t, a_t) - \zeta_{\pi_e}^{P,k} \right) \right| + \frac{L}{T}\sum_{k=1}^{K_2} \left| \sum_{e=1}^{E}\sum_{t=t_e}^{t_{e+1}-1} \left( \zeta_{\pi_e}^{P,k} - \zeta_{\pi_e}^{\tilde{P}_e,k} \right) \right| + C_1 \right)_{+} \tag{189}$$

$$\leq (C_3(T) + C_2(T) + C_1(T))_{+} \tag{190}$$

where Equation (186) follows from the convexity of the function $g$, and Equation (187) follows from the Lipschtiz continuity.

We bound $C_2(T) + C_3(T)$ similar to the analysis of $R(T)$ by

$$\tilde{\mathcal{O}}\left( T_M S\sqrt{\frac{A}{T}} + \frac{CT_M S^2 A}{(1-\rho)T} \right) \tag{191}$$

We focus our attention on bounding $C_1(T)$. For this, note that in Assumption 3.4 we assumed that the cost function $g$ is Lipschitz continuous and the gradients are bounded at all points. This implies for a bounded input domain the cost function is bounded. We assume that the upper bound of the cost is $g_\infty$. We now

obtain the bound on $C_1(T)$ as:

$$C_1(T) = \frac{1}{T} \sum_{e=1}^{E} T_e \left( g\left( \zeta_{\pi_e}^{\tilde{P}_e,1}, \cdots, \zeta_{\pi_e}^{\tilde{P}_e,K_2} \right) \right) \tag{192}$$

$$= \frac{1}{T} \sum_{e=1}^{E} T_e \left( g\left( \zeta_{\pi_e}^{\tilde{P}_e,1}, \cdots, \zeta_{\pi_e}^{\tilde{P}_e,K_2} \right) \right) \mathbf{1}\{T_e \geq T^{1/3}\}$$

$$+ \frac{1}{T} \sum_{e=1}^{E} T_e \left( g\left( \zeta_{\pi_e}^{\tilde{P}_e,1}, \cdots, \zeta_{\pi_e}^{\tilde{P}_e,K_2} \right) \right) \mathbf{1}\{T_e < T^{1/3}\} \tag{193}$$

$$\leq \frac{1}{T} \sum_{e=1}^{E} T_e \left( g\left( \zeta_{\pi_e}^{\tilde{P}_e,1}, \cdots, \zeta_{\pi_e}^{\tilde{P}_e,K_2} \right) \right) \mathbf{1}\{T_e \geq T^{1/3}\} + \frac{1}{T} \sum_{e=1}^{E} T^{1/3} g_\infty \tag{194}$$

$$\leq -\frac{1}{T} \sum_{e=1}^{E} T_e \epsilon_e \mathbf{1}\{T_e \geq T^{1/3}\} + \frac{1}{T} E T^{1/3} g_\infty \tag{195}$$

$$= -\frac{1}{T} \left( \sum_{e=1}^{E} T_e \epsilon_e \left( 1 - \mathbf{1}\{T_e < T^{1/3}\} \right) + E T^{1/3} g_\infty \right) \tag{196}$$

$$= -\frac{1}{T} \left( \sum_{e=1}^{E} T_e \epsilon_e + \frac{1}{T} \sum_{e=1}^{E} T_e \epsilon_e \mathbf{1}\{T_e < T^{1/3}\} + \frac{1}{T} E T^{1/3} g_\infty \right) \tag{197}$$

$$\leq -\frac{1}{T} \left( \frac{K}{4} \sqrt{T \log T} + \frac{1}{T} \sum_{e=1}^{E} T^{1/3} \kappa + \frac{1}{T} E T^{1/3} g_\infty \right) \tag{198}$$

$$= -\frac{K}{4T} \left( \sqrt{T \log T} + \frac{1}{T} E \kappa T^{1/3} + \frac{1}{T} E T^{1/3} g_\infty \right) \tag{199}$$

where Equation (194) follows from the bound on $g(\mathbf{x}) \leq g_\infty$. Equation (195) follows from following the conservative policy.

Thus, choosing an appropriate $K$, we can bound constraint violations by 0.

## H  Further Discussions

### H.1  Regarding Ergodicity in Assumption 3.1

Regarding the assumption on ergodicity in Assumption 3.1, we make two observations:

For MDPs with constraints, we note that for optimal policy to be stationary, the MDP has to be ergodic. For finite diameter MDPs, the optimal policy can be non-stationary. Consider an MDP with three states, left, middle, right, and two actions left, right. The left action keeps the state unchanged from left, or takes the agent left from middle and to middle from right. Similarly, the right action takes the agent to middle from left and to right from middle, or keep the agent to right. Since there exists different recurrent classes for different policies (For a policy which takes only left action, the recurrent class contains only left state. For policy which takes only right action, the recurrent class contains only right state), the MDP is non-ergodic. Further, the agent obtains a reward of +1 and cost of 0 on taking left action in left state and reward of 0 and cost of +1 on taking right action in the right state. A stationary policy provides an average reward of $(1/6, 1/6)$ as the agent stays in all three states with equal probability and takes either actions with equal probability in each state. Whereas, if the agent follows a a non-stationary optimal policy, the agent optimizes both rewards and cost with average reward of $(1/2, 1/2)$. Hence, the agent must stay in state left as much as the state right by only making minimal transitions via the middle state. Thus, the optimal policy is non-stationary for non-ergodic MDPs. This example is provided in detail by Cheung (2019).

The second observation is for finite diameter MDPs, where Chen et al. (2022) provided an algorithm which requires the knowledge of the time horizon $T$ and the span of the costs $sp_c$. We note that the two variables might not be known to the agent in advance. Further, the knowledge of the time horizon is required to divide

the time horizons to epochs of duration $O(T^{1/3})$ to obtain a regret bound of $O(T^{2/3})$. This particular epoch length is required to bound the bias-span of the MDP considered in the epoch. Finally, we note that the finite mixing time is also assumed in other works in constrained IH MDP (Singh et al., 2020).

We note that even if we use other exploration strategies, we will require the Bellman error analysis to analyze the stochastic policies. In this work, we perform an exploration and exploitation strategy by dividing the time horizon into epochs and then updating the policy in each epoch using the MDP model built using exploration done in previous epochs. The analysis of the regret will still need the impact of stochastic policies and thus the analysis approaching the paper will still be needed for any exploration strategy.

We also note that since the MDP is ergodic, exploration can be done with any policy and the agent does not need an optimistic MDP to explore. However, the agent wants to minimize the regret for the online algorithm, and hence it plays the optimal policy based on the MDP estimated/learned till time $t$. To do so the agent finds the optimistic policy or the policy which provides the highest possible reward in the confidence interval. Note that if agent agent could have played any policy and obtained same regret bound, a policy worse than the true MDP can also exist in the confidence interval and that would not give the same performance. In the following, we provide a simplified problem setup and algorithm to demonstrate that some policy may achieve large regret bound even with ergodic assumption.

Consider a simplified problem setup where $f(\lambda_\pi^P) = \lambda_\pi^P$ with no constraints. Note that this is the classical RL setup. Also consider an algorithm where the agent uses the estimated MDP without considering the confidence intervals. After every epoch, the agent solves for the optimal policy using the following optimization equation.

$$\max_{\rho(s,a)} \sum_{s,a} r(s,a)\rho(s,a) \tag{200}$$

with the following set of constraints,

$$\sum_{s,a} \rho(s,a) = 1, \quad \rho(s,a) \geq 0 \tag{201}$$

$$\sum_{a \in \mathcal{A}} \rho(s',a) = \sum_{s,a} \hat{P}_e(s'|s,a)\rho(s,a) \tag{202}$$

where $\hat{P}_e(\cdot|s,a)$ is the estimate for transition probability to next state given state action pair $(s,a)$ after epoch $e$. Let $\pi_e$ be the solution for optimization problem in Equation (200)-(202) for epoch $e$.

Now the regret $R(T)$, till time horizon $T$, is defined as

$$R(T) = T\lambda_{\pi^*}^P - \sum_{t=1}^{T} r(s_t,a_t) \tag{203}$$

$$= \sum_e \sum_{t=t_e}^{t_{e+1}-1} \lambda_{\pi^*}^P - \sum_e \sum_{t=t_e}^{t_{e+1}-1} r(s_t,a_t) \tag{204}$$

$$= \sum_e \sum_{t=t_e}^{t_{e+1}-1} \lambda_{\pi^*}^P \pm \sum_e \sum_{t=t_e}^{t_{e+1}-1} \lambda_{\pi_e}^P - \sum_e \sum_{t=t_e}^{t_{e+1}-1} r(s_t,a_t) \tag{205}$$

$$= \left( \sum_e \sum_{t=t_e}^{t_{e+1}-1} \lambda_{\pi^*}^P - \sum_e \sum_{t=t_e}^{t_{e+1}-1} \lambda_{\pi_e}^P \right) + \left( \sum_e \sum_{t=t_e}^{t_{e+1}-1} \lambda_{\pi_e}^P - \sum_e \sum_{t=t_e}^{t_{e+1}-1} r(s_t,a_t) \right) \tag{206}$$

$$= R_1(T) + R_2(T) \tag{207}$$

$R_2(T)$ is analysed in similarly to the regret analysis of the proposed UC-CURL algorithm. For $R_2(T)$, we obtain the following analysis:

$$R_1(T) = \sum_e \sum_{t=t_e}^{t_{e+1}-1} \left( \lambda_{\pi^*}^P - \lambda_{\pi_e}^P \right) \tag{208}$$

$$= \sum_e \sum_{t=t_e}^{t_{e+1}-1} \left( \left( \lambda_{\pi^*}^P - \lambda_{\pi_e}^{\hat{P}_e} \right) + \left( \lambda_{\pi_e}^{\hat{P}_e} - \lambda_{\pi_e}^P \right) \right) \tag{209}$$

Now, the second term, $\lambda_{\pi_e}^{\hat{P}_e} - \lambda_{\pi_e}^P$, in Equation (209) can be again analyzed using the Bellman error based analysis. We are primarily interested in the first term. Now, note that since the agent does not play the optimistic policy, $R_1(T)$ cannot be upper bounded by the optimal policy of the optimistic MDP in the confidence interval. For this, the average reward $\lambda_{\tilde{\pi}_e}^{\tilde{P}_e}$ satisfies $\lambda_{\tilde{\pi}_e}^{\tilde{P}_e} \geq \lambda_{\pi^*}^P$. For a posterior sampling algorithm, the optimal policy for the sampled MDP satisfies $\mathbb{E}[\lambda_{\tilde{\pi}_e}^{\tilde{P}_e}] = \mathbb{E}[\lambda_{\pi^*}^P]$. This two properties for the optimistic or posterior sampling algorithm contributes to the key parts in the analysis and design of the RL algorithms.

However, no such relationship can be established for $\lambda_{\pi^*}^P$, and $\lambda_{\pi_e}^{\hat{P}_e}$ and hence the first term of Equation (209) is not trivially upper bounded by 0. Further, for the optimal policy $\pi_e$ for the estimated MDP $\hat{P}_e$ can return a reward lower than the optimal policy $\pi^*$ on the true MDP $P$ or $\lambda_{\pi_e}^{\hat{P}_e} < \lambda_{\pi^*}^P$ resulting in a trivial $O(T)$ regret bound.

### H.2 Regarding Optimality

We note the work of (Singh et al., 2020) provided a lower bound of $\sqrt{DSAT}$, where $D$ is the diameter of the MDP, $A$, $S$ are the number of actions and states respectively and $T$ is the time horizon for which the algorithm runs. Based on the lower bound, the regret results presented in this work are optimal in $A$ and $T$. However, to obtain a tighter dependence on $S$ using tighter concentration inequalities for stochastic policies remain an open problem. Further, we note that reducing the dependence of $T_M$ to the diameter $D$ while also keeping the regret order in $T$ as $\tilde{O}(\sqrt{T})$ is an open problem.

## I Experiments with Fairness Utility and Constraints

We also evaluate the proposed algorithm on non-linear setup. We consider a scheduler allocating resources to two clients, $client_1, client_2$. At each time step, the scheduler allocates a resource to either of the clients. Hence, $\{client_1, client_2\}$ are 2 actions available to the scheduler. The client, on resource allocation, consumes the resource and obtains a reward. The reward depends on the state of the client. Each client can be in 4 possible states. Hence, there are 16 total possible system states. At every step a client stays in the same state with probability 0.625 and transitions to a different state with probability 0.125 of landing in any of the remaining 3 states.

The agent aims to maximize the proportional fairness among the two clients (Lan et al., 2010). The proportional fairness is used to quantitatively evaluate fairness in various networks scheduling systems such as wireless scheduling (Cui et al., 2019) and queuing (Wierman, 2011). We calculate proportional fairness as:

$$\sum_{i=\{1,2\}} \log \left( \sum_{s,a} \rho(s,a) r_i(s,a) \right) \tag{210}$$

where $i$ denotes the client index, $r_i(s,a)$ is the reward received by the client $i$ when the system is in state $s$ and takes action $a$, and $\rho(s,a)$ is the steady-state state-action distribution. The rewards are $r_i$ with respect to client state is presented in Table 3

Further, the first client is a high priority client and requires a minimum service level agreement (SLA) guarantee. Every time $client_1$ is denied the resource, the scheduler incurs a penalty of $-1$. Let $C$ denote the

Table 3: Transition probability of the queue system

| Client | Client State 1 | Client State 2 | Client State 3 | Client State 4 |
|--------|----------------|----------------|----------------|----------------|
| $client_1$ | 0.75 | 0.375 | 0.5 | 0.375 |
| $client_2$ | 0.25 | 0.5 | 0.75 | 1.0 |

SLA guarantee for $client_1$. Then the cost constraint can be written as:

$$-\sum_s \rho(s,a)\mathbf{1}_{\{a=client_2\}} \geq C \tag{211}$$

where $C$ is set to $-0.3$.

We evaluate the PS-CURL and UC-CURL algorithms on the scheduling system with $K = 1$. We evaluate both PS-CURL and UC-CURL algorithms for linear epochs and doubling epochs. We run 10 independent iterations of the algorithm for $T = 500000$ time steps. The mean values of the simulation results for the constrained setup are presented in Figure 2. The scheduler take about $100,000$ steps to converge at the optimal fairness value (Figure 2a) for PS-CURL algorithm where as the optimistic setup does not converges till $500,000$ steps. This is inline with the results of Section 6 where the posterior sampling algorithm converges the fastest. Further, note that optimistic algorithm is conservative with respect to constraints but it does not optimises for the fairness to satisfy the constraints.

We also present the system behavior in absence of constraints in Figure 2c and Figure 2d. For the unconstrained setup, we only evaluate the linear epoch setup of the PS-CURL algorithm which converges to the optimal fairness value at around $t = 50,000$ which is faster than the constrained setup. We also observe that the optimal fairness among the clients is higher when the scheduler is not required to guarantee any service level agreements.

Again, from the experimental evaluations, we observe that the proposed PS-CURL and UC-CURL algorithms can be used for systems with non-linear utilities and/or non-linear constraints.

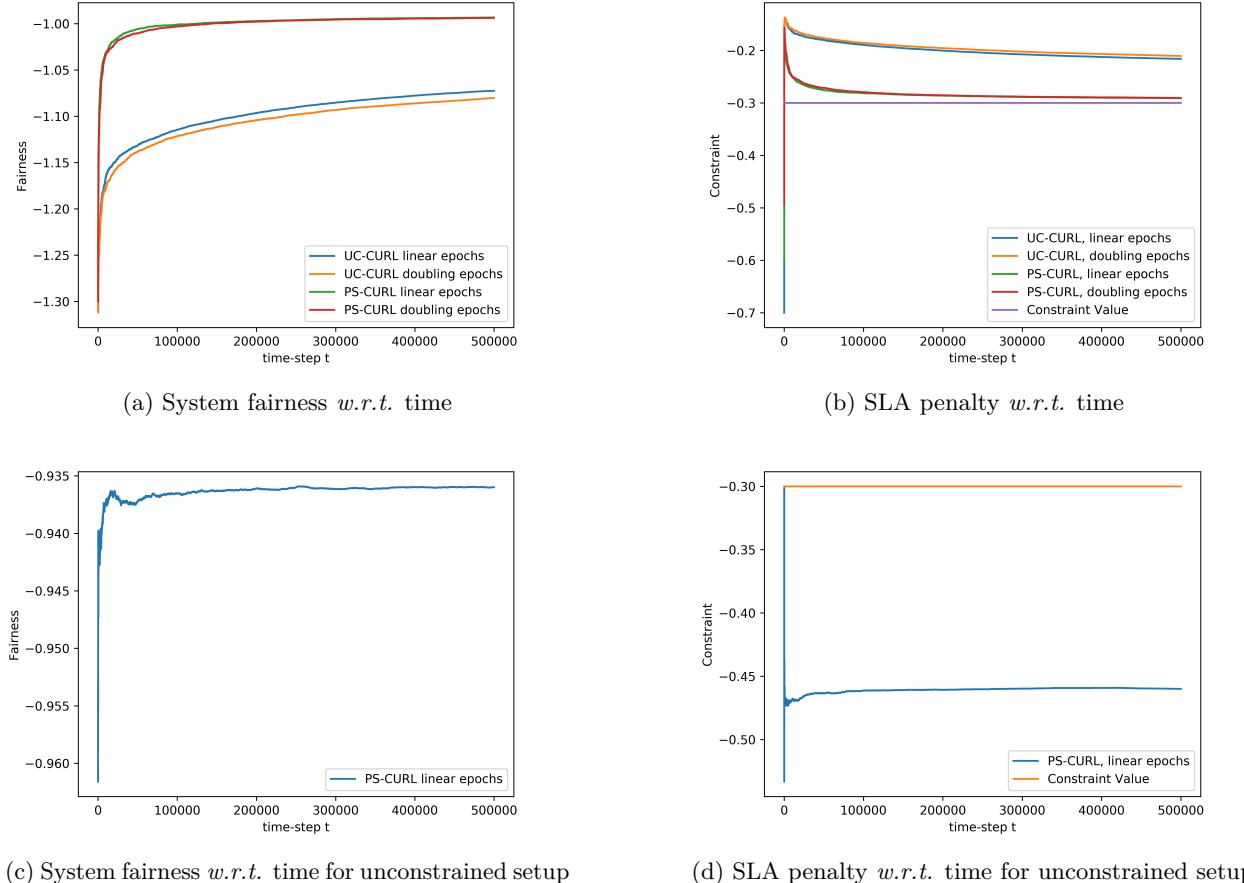

(a) System fairness *w.r.t.* time

(b) SLA penalty *w.r.t.* time

(c) System fairness *w.r.t.* time for unconstrained setup

(d) SLA penalty *w.r.t.* time for unconstrained setup

Figure 2: Performance of the proposed UC-CURL and PS-CURL algorithms on the scheduling example
.

