# OpenReview forum: "Concave Utility Reinforcement Learning with Zero-Constraint Violations"
_TMLR — Accepted by TMLR_

### Review · Reviewer_xsiA · 2022-09-29

**Summary Of Contributions:**

The paper studies the constrained reinforcement learning problem with general value functions in the infinite-horizon tabular setting. The authors employ the conservative approach from constrained optimization to propose a model-based RL algorithm for learning a constrained optimal policy. The authors prove that the proposed method has sublinear regret and zero constraint violation with high probability, under some assumptions on the underlying problem. The authors also show that the proposed method can be used for posterior sampling while enjoying a Bayesian regret. The authors also demonstrate the effectiveness of the proposed method by comparing it with other two methods in some experiments.

**Broader Impact Concerns:**

As mentioned, the paper studies a class of constrained RL problems that have the potential to address fairness issues. It is important to discuss this potential in depth: pros and cons of your model or method in real-world scenarios.

**Requested Changes:**

I provide some revision suggestions when weaknesses are discussed above.

Some other questions are given as follows.

(1) It is useful to provide a formal proof of the convexity of constraint Eq. (16), since it is not obvious.

(2) How do you choose the confidence bound in Eq. (18)? Is it optimal?

(3) In Algorithm 1, do you need to fix $T$? This is assumed by the theorem.

(4) Typos like 'Equation equation 11' and missing reference '(?)Lemma 1]osband2013more'

(5) In experiments, it seems the constrained value converges to some small positive constant. This might contradict the zero violation performance in theory. Can you comment the reason for this?

(6) It is helpful if the authors can test the proposed method in other practical applications, e.g., fairness?

(7) Grammatical errors need to be cleared.

**Strengths And Weaknesses:**

Strengths:

(1) The authors formulate a class of constrained reinforcement learning problems using the general value functions, instead of standard cumulative additive forms of reward functions. It is meaningful to study this class of RL problems since many practical RL objectives do not have the standard forms, e.g., regularized RL problems.

(2) The authors employ the conservative constrained optimization and the upper confidence bound of transitions to propose a UCRL type algorithm for the constrained RL problem with general value function. The authors provide a sublinear regret analysis that has some new features, e.g., the conservative offset can be used to limit the growth of the regret and constraint violation, and the performance gap of the optimistic policy under the optimistic and true models can be bounded by the Bellman errors.

(3) The authors provide an experiment to show how the proposed method can handle the constraints, which is useful.


Weaknesses:

(1) It is less discussed about the motivation of using general value functions in constrained reinforcement learning problems. It would be helpful to readers if the authors could discuss one or two typical examples in details, e.g., fairness.

(2) For the related work, it might be not fair to compare different algorithms from finite or infinite horizon setups. Although we can relate them conceptually, algorithms designed for these two setups seem to be different, especially how to design/assume online exploration. Instead of comparing all algorithms, it is useful to focus a class of algorithms in infinite horizon setup and make a comparison in more technical details.

(3) The ergodicity assumption (Assumption 3.1) is restrictive for the applicability of the proposed algorithm. It is more useful if the authors can weak this assumption or remove it, since the upper confidence bounds already encourage some level of exploration.

(4) In the regret bound, the probability depends on the length of horizon $T$. It might be confusing to call it high probability bound, since the probability can't be arbitrarily large in a way that is independent of the algorithm. It would be useful to remove the dependence on $T$ in the high probability.

(5) It is not very clear to me how posterior sampling works for your algorithms. It is useful to state the algorithm and regret in the paper since this is also an interesting part of the paper.

---

> ### Author Response · Authors · 2022-10-21
> **Regarding specific examples for the general setup**
>
> We thank the reviewer for pointing this out. We have added the fairness example in detail in the introduction. The additional details are:
>
> “Consider a scheduler which allocates a resource to $N$ users. Each user obtains some reward based on their current state. The goal of the scheduler is to maximize fairness among the users. However, there are certain preferred users for which some service level agreements (SLA) must be made. For this setup, the scheduler aims to find a policy which maximize the fairness while ensuring the SLA constraints of the preferred users are met. Note that, here, he objective is a non-linear concave utility in the presence of constraints on service level agreement”

---

> ### Author Response · Authors · 2022-10-21
> **Regarding comparisons with works in the finite horizon setup**
>
> We note the problem setup contains three important parts, non-linearity, constraints and infinite-horizon setup. We agree that for infinite horizon setup, the algorithms and analysis requires careful analysis which is because of:
>
> 1. For finite-horizon MDPs, the bias-span is bounded by the episode length and is known to the agent which is not true for the case of infinite horizon setup.
> 2. For infinite horizon MDPs, the policy switching cost has to be explicitly accounted for which is not the case with finite horizon setup as the MDP restarts after every episode.
>
> However, for the completeness of the problem setup from the lens of non-linearity and constrained RL, we introduced the works in the field of finite horizon as well. Further, we have acknowledged the initial works in the area of RL which considers zero-constraint violations and concave objectives.

---

> ### Author Response · Authors · 2022-10-21
> **Regarding ergodicity assumption**
>
> Regarding ergodicity, we make two observations:
>
> 1. For MDPs with constraints, we note that for optimal policies to be stationary, the MDP has to be ergodic. For finite diameter MDPs, the optimal policies can be non-stationary. Consider an MDP with three states, {left, middle, right}, and two actions {left, right}. The left action keeps the state unchanged from left, or takes the agent left from middle and to middle from right. Similarly, the right action takes the agent to middle from left and to right from middle, or keeps the agent to right. Further, the agent obtains a reward of +1 and cost of 0 on taking left action in left state and reward of 0 and cost of +1 on taking right action in the right state. Thus, to optimize both rewards and cost, the agent must stay in state left as much as the  state right by only making minimal transitions via the middle state. This example is provided in detail by Cheung W.C., 2019.
>
> 2. The second observation is for finite diameter MDPs, Chen et al provided an algorithm which requires the knowledge of the time horizon $T$ and the span of the costs $sp_c$. We note that the two variables might not be known to the agent in advance. Further, the knowledge of the time horizon is required to divide the time horizons to epochs of duration $O(T^{1/3})$. This particular epoch length is required to bound the bias-span of the MDP considered in the epoch. Further, even with this knowledge, the analysis obtains a loose regret bound of $O(T^{2/3})$. Finally, we note that the finite mixing time is also assumed in other works in constrained IH MDP (Singh et al., 2022).
>
>
> Based on these observations, we note that the ergodicity assumption is indeed required to obtain a tighter regret bound.
>
> Cheung, Wang Chi. "Regret minimization for reinforcement learning with vectorial feedback and complex objectives." Advances in Neural Information Processing Systems 32 (2019).
> Singh, R., Gupta, A., & Shroff, N. (2022). Learning in Constrained Markov Decision Processes. IEEE Transactions on Control of Network Systems.
> Chen, L., Jain, R., & Luo, H. (2022). Learning Infinite-Horizon Average-Reward Markov Decision Processes with Constraints. ICML 2202.

---

> ### Author Response · Authors · 2022-10-21
> **Regarding the theorem/algorithm dependent on time horizon T**
>
> We note that the algorithm does not assume any dependence on T, and is thus independent of T.
>
> Further, the regret till time $T$ has to be a function of T, its dependence is used in the regret bound analysis. Thus, the analysis is a function of T, while the algorithm is not. Further, we can make the high probability result independent of $T$ by simply replacing $1-1/T^{5/4}$ with $1-(\kappa/T)^{5/4} \ge 1-\delta$ for some $\kappa \in (0, 1]$. This will simply increase the regret by a logarithmic factor of $\log (1/\kappa)$, with new regret being:
> $O(\frac{1}{\delta}LdT_{mix}S\sqrt{\frac{A\log{T/\kappa}}{T}})$

---

### Review · Reviewer_yAcd · 2022-09-30

**Summary Of Contributions:**

This work studies reinforcement learning with constraints, and is the first to consider this setting with nonlinear constraints and objectives in the infinite horizon setting. At each step, the learner receives a reward signal and $d$ constraint signals. The goal is to then maximize a (concave) function of the average reward, while ensuring that the average constraint signals satisfy a convex constraint. They propose an optimism-style algorithm and show that it provably achieves $O(1/\sqrt{T})$ average regret and the final constraint signal averages satisfy the desired constraint with high probability.

**Broader Impact Concerns:**

None.

**Requested Changes:**

- It is not clearly stated what the goal is. From reading equation (3), it is unclear whether the agent simply wants to find a policy with maximal reward satisfying the constraints (a PAC problem), or solve it in an online manner. This becomes clear by section 5, but clearly stating the objective in section 3 would help clarify things.
- In general the standard RL setup considers linear rewards (and linear constraints, in the constrained setting). Several examples are given to justify the study of nonlinear functions of the rewards and constraints but, given that it is somewhat non-standard and that this is the primary novelty of the work, more justification and discussion would be helpful.
- As mentioned above, (Chen et al., 2022) also consider the weakly communicating case (I.e. removing the ergodicity assumption). Obtaining a result in this setting would be ideal, but at least some discussion of this setting should be given.
- The setting of $K$ given in Theorem 5.6 depends depends on $T_M, C$, and $\rho$, which the learner in general may not know. It would be helpful to state how the regret and constraint violations scale if these parameters are not known and $K$ is therefore chosen differently.
- There were numerous minor typos and incorrect word uses throughout the paper. For example, in the last paragraph on page 1, the sentence beginning “Much recently” should begin “More recently”. There were many other similar instances (too many for me to list here), or places where “the” or other minor words were dropped. The main idea was still conveyed in every case I found, but the number of typos was very distracting. As such, I think it’s important that these are fixed for the final version.

In addition, there were several missing or unclear definitions throughout that should be fixed. In particular:
- The definition of the MDP should contain an initial state distribution.
- It is never spelled out what the acronym “CMDP” means.
- $\pi*$ is never defined.
- The definition of the mixing time should have a max over $s$ and $s’$ as well I believe.
- It is stated in words what $V_\gamma^{\pi,P}(s)$ is but this should be defined mathematically.
- $V_\gamma^{\pi,P}(s;i)$ is never defined.
- $P_{\pi,s}^t$ is defined in words but it would be helpful to also define mathematically.
- In the definition of $C(T)$, I believe the subscript on the final $c$ term should be $c_d$ not $c_1$.
- In the experiments section, the PS-CURL algorithm is referenced but it is never stated in the text what algorithm this is (I assume it is the algorithm described in section 5.1 but this should be clarified).
While several of these terms might be obvious and standard, I believe it is important that they are clearly spelled out.


**Strengths And Weaknesses:**

Strengths:
- There has been a large amount of work over the last few years on constrained RL. This work considers a novel constrained RL setting—infinite horizon with nonlinear objective and constraints—and obtains a guarantee which, at least in the horizon dependence, appears to be optimal.
- As far as I can tell the results are technically correct.
- Simulation results demonstrate that a posterior sampling version of the algorithm achieves state-of-the-art performance.

Weaknesses:
- The result feels somewhat incremental. My understanding is that the primary novelty is considering nonlinear constraints in the infinite-horizon setting. Infinite-horizon constrained RL with linear constraints and finite horizon with nonlinear constraints have both been studied before, so the only real novelty is combining these.
- In addition, the techniques and algorithm feel largely standard.
- The ergodicity assumption seems rather strong, and is likely not necessary ((Chen et al., 2022) remove this assumption in the linear case, with somewhat weaker guarantees).
- No discussion of optimality is given. It seems the $T$ dependence is optimal, but it’s not clear the dependence on the remaining parameters is optimal (I would guess it is not—for example you would expect a $\sqrt{S}$ instead of the obtained $S$ dependence).
- The exposition could be cleaned up somewhat (see comments below).

---

> ### Author Response · Authors · 2022-10-21
> **Regarding Incremental results**
>
> For Infinite horizon, our results are better than the current existing results. In the literature of constrained reinforcement learning for infinite horizon, only two prior works exist. The first work is by Singh et al., 2022 which uses forced exploration and obtains a regret bound of $O(T_{mix}/{(1-\rho)}T^{2/3})$. The second work of by Chen et al., 2022 provides an algorithm with confidence intervals  on the value function and also considers extended MDPs for value iteration to obtain a regret bound of $O(T_{mix}/{(1-\rho)^2}\sqrt{S^3AT})$.
>
> Compared to these works, we also consider the general problem of concave objective and convex constraints. Further, our analysis obtains a better regret guarantee of $O(T_{mix}/(1-\rho)S\sqrt{AT}).
>
> We improve on the result of Singh et al., by showing that the bias span of the optimistic/sampled MDP is bounded by $T_{mix}$ and hence the optimal policy for the optimistic/sampled MDP encourages sufficient exploration. Further, our bound on the bias-span of $T_{mix}$ is tighter compared to the bound obtained by Chen et al of $T_{mix}/(1-\rho)$ which improves a factor of $1/(1-\rho)$. Further, as compared to the analysis of Chen et al., we only consider the confidence intervals for the transition probability estimates which allows for the improvement of $O(\sqrt{S})$.
>
> Moreover, our analysis allows for analysing Bayesian regret bound for the posterior sampling based algorithms by obtaining feasibility of the optimization equation for the sampled policy
>
> Singh, R., Gupta, A., & Shroff, N. (2022). Learning in Constrained Markov Decision Processes. IEEE Transactions on Control of Network Systems.
> Chen, L., Jain, R., & Luo, H. (2022). Learning Infinite-Horizon Average-Reward Markov Decision Processes with Constraints. ICML 2202.

---

> ### Author Response · Authors · 2022-10-21
> **Regarding standard algorithm and analysis techniques**
>
> We agree that the algorithm is standard. But, the analysis is novel on accounts of two parts. We use a novel Bellman error based analysis which allows us to analyse stochastic policies easily by only considering confidence intervals for the transition probability vectors. The second novelty is the tighter bound on the bias-span (compared to Chen et al., 2022) of the optimal policies for sampled/optimistic MDP.
>
> Together, these lemmas allow us to analyse the optimistic and posterior sampling algorithm. To the best of our knowledge, we note that our work is the first work to provide regret guarantees for the posterior sampling algorithm in the general setup of concave utilities and convex constraints.

---

> ### Author Response · Authors · 2022-10-21
> **Regarding ergodicity assumption**
>
> Regarding ergodicity, we make two observations:
>
> 1. For MDPs with constraints, we note that for optimal policies to be stationary, the MDP has to be ergodic. For finite diameter MDPs, the optimal policies can be non-stationary. Consider an MDP with three states, {left, middle, right}, and two actions {left, right}. The left action keeps the state unchanged from left, or takes the agent left from middle and to middle from right. Similarly, the right action takes the agent to middle from left and to right from middle, or keeps the agent to right. Further, the agent obtains a reward of +1 and cost of 0 on taking left action in left state and reward of 0 and cost of +1 on taking right action in the right state. Thus, to optimize both rewards and cost, the agent must stay in state left as much as the  state right by only making minimal transitions via the middle state. This example is provided in detail by Cheung W.C., 2019.
>
> 2. The second observation is for finite diameter MDPs, (Chen et al., 2022) provided an algorithm which requires the knowledge of the time horizon $T$ and the span of the costs $sp_c$. We note that the two variables might not be known to the agent in advance. Further, the knowledge of the time horizon is required to divide the time horizons to epochs of duration $O(T^{1/3})$. This particular epoch length is required to bound the bias-span of the MDP considered in the epoch. Further, even with this knowledge, the analysis obtains a loose regret bound of $O(T^{2/3})$. We do not assume such knowledge of the time horizon $T$ and the span of the costs $sp_c$ in this paper. Finally, we note that the finite mixing time is also assumed in other works in constrained IH MDP (Singh et al., 2022).
>
> Thus, we believe that analyzing the regret for ergodic MDP is novel, and removing such assumption is a possible future direction. Further, we have added the discussion in the Appendix of the revision.
>
> Cheung, Wang Chi. "Regret minimization for reinforcement learning with vectorial feedback and complex objectives." Advances in Neural Information Processing Systems 32 (2019).

---

> ### Author Response · Authors · 2022-10-21
> **Discussion regarding (sub)-optimality**
>
> We agree that S dependence may not be optimal. In the prior works, Bernstein inequalities have been used instead of Hoeffding to get $\sqrt{S}$, while it is left as an open problem if such techniques can be used in this work with stochastic policies. Further, we are optimal in orders of A and S. We have mentioned the lower bound for linear case, and have described the optimality in orders of T and A in the revision.

---

### Review · Reviewer_7s2j · 2022-10-08

**Summary Of Contributions:**

The paper studies tabular infinite horizon concave utility RL with concave with convex constraints. A model-based RL approach was proposed, with zero constraint violation guarantees. Regret analyses have been developed for both optimistic methods and Thompson sampling methods.

1. How strong and necessary the assumptions of knowing the reward and cost functions (Assumption 3.2) are? This seems to be not exactly the RL setting.

2. Why Assumption 3.1 is necessary? Is it assumed to hold for ANY pi (and with finite mixing time)? If so, why we even need to explore, using optimistic methods, to get a good regret analysis? I might be a bit confused about this assumption, compared to the literature on average-reward MDP.

3. Can we elaborate more about Assumption 3.7? what is C, and what is the intuition of  being larger than the constants?

4. Does Eq (14) have a typo?

5. Can the authors give more examples about the usefulness and expressiveness of the mode of general utility with general constraints? It seems that even in the simulations, the comparison was with the recent average-reward MDP setting (not the general utility one). It would be better to make use of the developed methods in other general examples.

6. How optimal is the upper bound? There is an S dependence (instead of sqrt S), is this optimal? Since this is a new and more general setting, it would be better to elaborate more about this point, or ideally give some lower bound.

7. The writing of the paper needs a bit more work. Typos: 1). Second paragraph on page 2, in the beginning "in past..." does not read well. 2). First line on page 2, "bound" -> "bounds". 3). When referring to references, sometimes the present, but sometimes the past tenses are used. it's better to make it consistent. 4). Some equations are not ended with period, or sometime it should be, but ended with a comma. 5). Two lines after eq. (3), "let ... denotes" -> "let ... denote".

In sum, the paper deals with a more general RL setting, with some interesting theoretical developments. However, the paper might still need to be improved in order to clear the bar of acceptance.

**Requested Changes:**

1. It would be good if the authors could elaborate and justify more about the assumptions that I have mentioned above.
2. A thorough and careful grammar and writing check is necessary.
3. Discussion about the technical novelty is necessary.
4. Discussion on the optimality of the upper bound would be ideal.

**Strengths And Weaknesses:**

Strengths: the paper's technique is solid, and the setting is general.

Weakness: some of the assumptions might require further discussion; the writing needs to be improved; the algorithms and analyses seem standard, and their novelty might need to be discussed more.

---

> ### Author Response · Authors · 2022-10-22
> **Regarding knowledge of reward/costs (Assumption 3.1)**
>
> We note that the assumption 3.2 of known rewards and cost is not required. If the setup does not satisfy the assumption 3.2, we require a few minor changes in the algorithm and the analysis.
>
> For the algorithm, we require confidence intervals around the estimated rewards and estimated costs as well. This introduces 2 additional constraints for rewards as $r - \hat{r} \le confidence$ and $\hat{r}-r \le confidence$ and similarly 2 additional constraints per cost estimate for the optimistic algorithm. For the posterior algorithm, the cost and rewards will be sampled from the updated posterior.
>
> Let $\tilde{r}$ be the sample or the solution for reward and $\tilde{c}_i$ be the sample or the solution for the cost $i$.
> For the analysis, we require the following changes:
>
> 1. We require to introduce a new variable $\tilde{Q}_\gamma^{\pi_e, \tilde{P}_e} (s,a) = \tilde{r}(s,a) + \gamma \sum \tilde{P}(\cdot|s,a)\tilde{V}^{\pi_e, \tilde{P}_e}_\gamma(\cdot)$ to account for the fact that the optimistic/sampled MDP can have a different reward compare to the true MDP.
>
> 2. The major changes comes in Lemma D.3 where we consider $\tilde{Q}^{\pi_e, \tilde{P}_e}_\gamma (s,a) - r(s,a) -\gamma\sum P(\cdot|s,a)\tilde{V}^{\pi_e, \tilde{P}_e}_\gamma(\cdot)$. Now the $\tilde{r}(s,a)-r(s,a)$ term does not cancel but propagates through the analysis. Since the sampled/optimistic MDP are close enough in the confidence intervals, we know that $\tilde{r}(s,a)-r(s,a) \le confidence \  diameter \le \sqrt{\frac{\log \delta}{N_e(s,a)}}$. This is an added term in RHS of Equation 92.
>
> 3. The final change comes accounting the updated Lemma D.3 and summing the term using the Proposition 19 of Jaksch et al., 2010
>
> In conclusion, the removal of the assumption does not impact the algorithm or the analysis significantly, however it unnecessarily complicates the analysis with additional terms and hence to keep the analysis simple, we used the Assumption 3.2
>
> Auer, P., Jaksch, T., & Ortner, R. (2008). Near-optimal regret bounds for reinforcement learning. Advances in neural information processing systems, 21.

---

> ### Author Response · Authors · 2022-10-22
> **Regarding Ergodicity Assumption (Assumption 3.1)**
>
> Regarding ergodicity, we make two observations:
>
>
> For MDPs with constraints, we note that for optimal policy to be stationary, the MDP has to be ergodic. For finite diameter MDPs, the optimal policy can be non-stationary. Consider an MDP with three states, {left, middle, right}, and two actions {left, right}. The left action keeps the state unchanged from left, or takes the agent left from middle and to middle from right. Similarly, the right action takes the agent to middle from left and to right from middle, or keep the agent to right. Further, the agent obtains a reward of +1 and cost of 0 on taking left action in left state and reward of 0 and cost of +1 on taking right action in the right state. Thus, to optimize both rewards and cost, the agent must stay in state left as much as the  state right by only making minimal transitions via the middle state. This example is provided in detail by Cheung W.C., 2019. We will add the result with reference in the revision.
>
> The second observation is for finite diameter MDPs, Chen et al. provided an algorithm which requires the knowledge of the time horizon $T$ and the span of the costs $sp_c$. We note that the two variables might not be known to the agent in advance. Further, the knowledge of the time horizon is required to divide the time horizons to epochs of duration $O(T^{1/3})$ to obtain a regret bound of $O(T^{2/3})$. This particular epoch length is required to bound the bias-span of the MDP considered in the epoch. Finally, we note that the finite mixing time is also assumed in other works in constrained IH MDP (Singh et al., 2022).
>
> Regarding the comment for exploration, we still need to explore using a stationary policy, which is enabled by dividing the time in epochs. The analysis of the regret will still need the impact of stochastic policies and thus the analysis approaching the paper will still be needed for any exploration strategy.
>
> Further, regarding any policy will perform exploration: We agree that exploration can be done with any policy. However, we want to minimize the regret for the online algorithm, hence we aim to play the optimal policy based on the MDP estimated/learned till time $t$. To do so we aim to find the optimistic policy or the policy which provides the highest possible reward in the confidence interval. Note that if agent agent could have played any policy and obtained same regret bound, the worst possible policy can also exist in the confidence interval and that would not give the same performance.
>
> Based on these reasons, we believe that analyzing the regret for ergodic MDP is novel. Further, removing ergodicity assumption while still obtaining a $O(\sqrt{T})$ regret bound possible future direction.
>
> Cheung, Wang Chi. "Regret minimization for reinforcement learning with vectorial feedback and complex objectives." Advances in Neural Information Processing Systems 32 (2019).
>
> Singh, R., Gupta, A., & Shroff, N. (2022). Learning in Constrained Markov Decision Processes. IEEE Transactions on Control of Network Systems.
>
> Chen, L., Jain, R., & Luo, H. (2022). Learning Infinite-Horizon Average-Reward Markov Decision Processes with Constraints. ICML 2202.

---

> ### Author Response · Authors · 2022-10-22
> **Regarding Assumption 3.7 (Existence of Slater's constant), C and the intuition of being larger than constants**
>
> In Assumption 3.7, we assume the existence of policy $\pi$ such that the constraint value $g\left(\zeta_{\pi}^P(1), \cdots, \zeta_{\pi}^P(d)\right) \le -\delta$. Here $\delta > LdST_M\sqrt{(A\log T)/T} + (CSA\log T)/(T(1-\rho)) $. We note that the Slater's constant has to be large enough to provide sufficient slack for the agent to course correct. Further, note that the Slater's constant is inversely proportional to the time horizon. This implies that the larger the time horizon available to the agent, the agent can stay inside the feasible zone for a longer duration to compensate for the constraints violated in the initial phases of the time horizon.
>
> Further, $C$ is not any arbitrary constant but the constant from Assumption 3.1 specifying the distance between the stationary distribution for any policy $\pi$ and its $t$-step distribution.

---

> ### Author Response · Authors · 2022-10-22
> **Usecases/Usefullness of a general utility model**
>
> We thank the reviewer for pointing this out. We have added the fairness example in detail in the introduction. The additional details are:
>
> “Consider a scheduler which allocates a resource to  users. Each user obtains some reward based on their current state. The goal of the scheduler is to maximize fairness among the users. However, there are certain preferred users for which some service level agreements (SLA) must be made. For this setup, the scheduler aims to find a policy which maximize the fairness while ensuring the SLA constraints of the preferred users are met. Note that, here, he objective is a non-linear concave utility in the presence of constraints on service level agreement”
>
> Further, in the simulations, we compared with the existing work in the area of infinite horizon constrained RL setup. The existing work in that area considers only linear setups.

---

> ### Author Response · Authors · 2022-10-22
> **Regarding optimality of the algorithm**
>
> We note that the work of Singh et al., provided a lower bound for the setup with linear rewards and constraints. Their lower bound is $O(\sqrt{DSAT})$ where $D$ is the diameter of the MDP. Based on that lower bound, our work is looser by a factor of $O(\sqrt{S}$. Further, our regret bound depends on the mixing time and assumes ergodicity whereas the lower bound is dependent on the diameter. Hence, our algorithm is also makes use of a tighter assumption as well.

---

### Decision · Action_Editors · 2022-11-25

**Recommendation:** Accept with minor revision

**Comment:**

Reviewers are generally positive about the work, which gives solid contributions in an interesting setting. Even though the algorithm and analysis are built on previous work, the results are useful additions to the literature. The author responses clarified several questions raised in the reviews, and made further improvements such as adding a discussion of the ergodicity assumption and proofreading. The work is now ready for acceptance.

A few minor suggestions:

* When a citation is used as a noun, use \citet instead of \citep. So "the discussion in (Altman & Schwartz, 1991)" would become "the discussion in Altman & Schwartz (1991)".

* Make the language in assumption A.1 precise. "The MDP M is ergodic, or ..." Should "or" be "and" instead?

* There is no need to start a new page for a new section in the appendix.

**Audience:**

Yes.

**Claims And Evidence:**

The claims are supported by theoretical analysis and simulation studies.